# Proteomic profiling reveals CDK6 upregulation as a targetable resistance mechanism for lenalidomide in multiple myeloma

Yuen Lam Dora Ng [1,10], Evelyn Ramberger [1,2,3,10], Stephan R. Bohl[4,5], Anna Dolnik[1], Christian Steinebach [6], Theresia Conrad[7], Sina Müller[1], Oliver Popp[2], Miriam Kull[4], Mohamed Haji[2], Michael Gütschow[6], Hartmut Döhner[4], Wolfgang Walther[8], Ulrich Keller [1,3,8,9], Lars Bullinger[1,3,9], Philipp Mertins [2,9✉] & Jan Krönke [1,3✉]

The immunomodulatory drugs (IMiDs) lenalidomide and pomalidomide are highly effective treatments for multiple myeloma. However, virtually all patients eventually relapse due to acquired drug resistance with resistance-causing genetic alterations being found only in a small subset of cases. To identify non-genetic mechanisms of drug resistance, we here perform integrated global quantitative tandem mass tag (TMT)-based proteomic and phosphoproteomic analyses and RNA sequencing in five paired pre-treatment and relapse samples from multiple myeloma patients. These analyses reveal a CDK6-governed protein resistance signature that includes myeloma high-risk factors such as TRIP13 and RRM1. Overexpression of CDK6 in multiple myeloma cell lines reduces sensitivity to IMiDs while CDK6 inhibition by palbociclib or CDK6 degradation by proteolysis targeting chimeras (PROTACs) is highly synergistic with IMiDs in vitro and in vivo. This work identifies CDK6 upregulation as a druggable target in IMiD-resistant multiple myeloma and highlights the use of proteomic studies to uncover non-genetic resistance mechanisms in cancer.

[1] Department of Hematology, Oncology and Cancer Immunology, Charité - Universitätsmedizin Berlin, corporate member of Freie Universität Berlin and Humboldt-Universität zu Berlin, Berlin, Germany. [2] Proteomics Platform, Max Delbrück Center for Molecular Medicine, Berlin, Germany. [3] German Cancer Consortium (DKTK) partner site Berlin and German Cancer Research Center (DKFZ), Heidelberg, Germany. [4] Department of Internal Medicine III, Ulm University Hospital, Ulm, Germany. [5] Department of Medical Oncology, Dana-Farber Cancer Institute, Harvard Medical School, Boston, MA, USA. [6] Department of Pharmaceutical & Medicinal Chemistry, Pharmaceutical Institute, University of Bonn, Bonn, Germany. [7] Experimentelle Pharmakologie & Onkologie (EPO) Berlin-Buch GmbH, Berlin, Germany. [8] Experimental and Clinical Research Center, Charité Universitätsmedizin Berlin and Max-Delbrück-Center for Molecular Medicine, Berlin, Germany. [9] Berlin Institute of Health (BIH), Berlin, Germany. [10] These authors contributed equally: Yuen Lam Dora Ng, Evelyn Ramberger. ✉email: philipp.mertins@mdc-berlin.de; jan.kroenke@charite.de

Multiple myeloma is a genetically heterogeneous malignancy of plasma cells. The immunomodulatory imide drugs (IMiDs) lenalidomide and pomalidomide are a mainstay in treating multiple myeloma[1]. Although the combination of IMiDs with other drugs like proteasome inhibitors, antibodies, corticosteroids, and high-dose chemotherapy can induce remissions in most patients, almost all patients eventually relapse due to acquired resistance of the multiple myeloma cells to one or several of the drugs[1]. IMiDs bind to cereblon (CRBN), that together with DDB1, CUL4A, and ROC1 forms the CRBN-CRL4 E3 ubiquitin ligase and modulate the substrate specificity of the enzyme[2]. This leads to ubiquitination and proteasomal degradation of the lymphoid transcription factors Ikaros (IKZF1) and Aiolos (IKZF3), which regulate expression of other genes such as *IRF4* and *MYC*, and are essential for the proliferation and survival of multiple myeloma cells[3–7]. Pomalidomide is the most potent of the approved IMiDs, both in regard to IKZF1 and IKZF3 degradation, as well as clinical activity, and is therefore a preferred treatment for relapsed multiple myeloma[3,8]. Sequencing studies in relapsed multiple myeloma and functional screens identified acquired genetic alterations in members of the CRBN-CRL4 E3 ligase complex that completely abrogate lenalidomide and pomalidomide activity as an IMiD-specific resistance mechanism in 10–20% of relapsed patients[9–13]. In single cases, IMiD-resistance was found to be caused by *IKZF1* mutations at the critical degron region, which blocks IMiD-induced IKZF1 degradation[10]. DNA sequencing of heavily pre-treated multiple myeloma patients identified additional recurrent mutations and aberrations enriched at relapse including homozygous inactivation of tumor-suppressor genes *TP53*, *RB1*, *FAM46C*, *BIRC3*, *TRAF3*[14–16]. However, only few of these aberrations have been directly linked to the activity of IMiDs or other drugs used in multiple myeloma[17]. Furthermore, inactivating mutations in tumor-suppressor genes are in general not amenable to pharmacologic interventions. Gene expression profiling (GEP) has found an enrichment of the GEP70 prognostically high-risk signature in relapsed cases[15,18,19]. Like genetic alterations, this signature was not associated with a specific treatment. In aggregate, these previous studies imply that genetic alterations alone do not fully explain the occurence of drug resistance in multiple myeloma. In addition, protein abundance and activity frequently cannot be inferred from RNA expression analyses due to post-transcriptional regulation mechanisms in general[20], and in cancer in particular, due to complex compensation effects of genetic alterations on the protein level[21–23]. Proteomic profiling in cell lines and pooled patient samples has been successfully applied to study drug resistance mechanisms in patients with hematological disorders such as FLT3 inhibitor-resistant acute myeloid leukemia[24] or bortezomib refractory multiple myeloma[25,26].

Here, we apply quantitative proteomic analyses in paired, longitudinal primary multiple myeloma samples and identify CDK6 upregulation as a non-genetic resistance mechansim for IMiDs in multiple myeloma that can be overcome by pharmacologic intervention.

## Results

**Quantitative proteomic analysis identifies deregulated protein abundance levels in relapsed multiple myeloma**. To identify deregulated proteins in relapsed multiple myeloma, five patients with available longitudinal bone marrow samples were included in our study. Patients progressed during ($N = 4$) or shortly after ($N = 1$) lenalidomide-comprising treatment (Supplementary Fig. 1). Paired bone marrow samples obtained pre-treatment and at relapse were lysed, trypsin digested, labeled with isobaric tandem mass tags (TMT) and analyzed with quantitative mass spectrometry (Fig. 1A).

In total, we quantified 6094 proteins with at least two peptides without missing values across all samples (Supplementary Data 1). Using an FDR cutoff of 0.10, we found 130 proteins upregulated and 228 proteins downregulated in the relapse vs. pre-treatment samples (Fig. 1B). The top six upregulated proteins ranked by FC and FDR were TRIP13, RRM1, NCAPD2, NCAPH, MORF4L1, and CDK6 (Table 1), and the six most downregulated proteins were UPRT, DNAJC1, FCRL2, AUH, HYI and HID1 (Table 2). Although all five patients received lenalidomide and dexamethasone during their treatment, we did not detect changes in proteins involved in the mechanism of IMiDs (CRBN, DDB1, IKZF1, IKZF3, IRF4, BSG)[2,3,27] or the glucocorticoid receptor NR3C1 (Supplementary Fig. 2A). However, several of the top upregulated proteins have been previously found to be implicated in multiple myeloma and were further pursued: the ATPase TRIP13 promotes progression of B-cell malignancies[28] and is part of the validated GEP70 and GEP5 multiple myeloma gene expression high-risk signature[29,30]; RRM1 plays a role in DNA synthesis and repair, is essential for multiple myeloma cell proliferation and its expression is linked to shorter survival[31]; the cell-cycle regulator CDK6 is dysregulated in multiple myeloma and CDK6 inhibitors have shown activity in multiple myeloma in early clinical trials[32–34]. Protein analyses by western blot in an independent cohort with four samples obtained from patients at first diagnosis and nine samples from patients obtained at relapse confirmed that CDK6, TRIP13, and RRM1 proteins are more frequently detected in the relapse samples as compared to the pre-treatment samples (Fig. 1C, Supplementary Fig. 1).

**Comparison of proteome, phosphoproteome, and RNA expression analyses**. To determine whether the differential expression of the proteins was accompanied by changes in RNA expression levels, we performed RNA sequencing of the five paired samples analyzed by proteomics (Supplementary Data 2). The top upregulated RNAs at relapse versus pre-treatment samples were *ADGRG3*, *FCAR*, *CAMP* and *G0S2*. Only two of the downregulated RNA transcripts, *PAIP2B* and *ZBTB20* had an FDR below 0.1 (Supplementary Fig. 3A).

The general correlation of protein and RNA expression changes between pre-treatment and relapse among all protein/RNA pairs was weak with a median Pearson correlation coefficient (PCC) of 0.34 with a high degree of variation (range −1 to 1) (Supplementary Fig. 3B). Of the top upregulated proteins, the mitosis regulatory protein TRIP13 showed the highest level of correlation for RNA/protein expression (PCC = 0.84), followed by NCAPH (0.84) and NCAPD2 (0.67). RRM1 and CDK6 had an RNA to protein correlation of 0.6 and 0.39, respectively (Fig. 1D).

In addition to analyzing the global proteome, we also performed an immobilized metal affinity chromatography (IMAC) phosphopeptide enrichment with 9 of the 10 samples. We detected 24,796 phosphopeptides derived from 5698 proteins (Supplementary Fig. 3C, Supplementary Data 1). In total, 134 phosphopeptides passed the 0.12 FDR significance cutoff. The majority of the proteins, that the significant phosphopeptides originated from, were also detected in the global proteome analysis of patient samples (92 out of 112). However, only 15 of the significant phosphopeptides belonged to proteins that were also significantly regulated on the global protein level.

The complementary nature of the different datasets was also reflected by highly significant single sample gene set enrichment analysis (ssGSEA) signatures observed in the phosphoproteomic data and, to a lesser extent, in the proteomic data (Supplementary Fig. 3D). SsGSEA revealed upregulation of cell cycle-

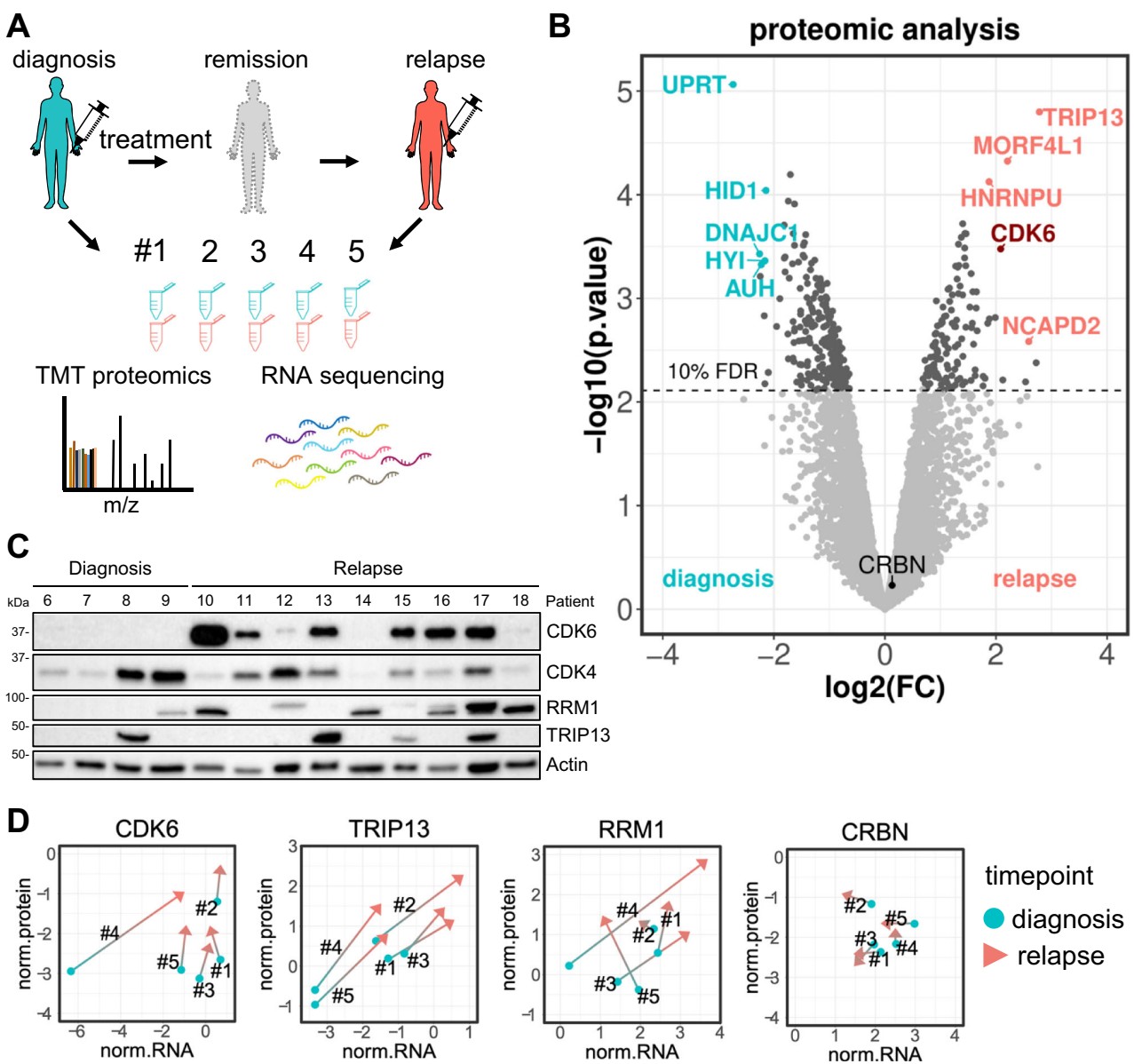

**Fig. 1 Identification of CDK6 protein upregulation in relapsed multiple myeloma patients. A** Bone marrow samples of five multiple myeloma patients were obtained at diagnosis and at relapse. Samples were subjected to TMT-based quantitative proteomic analysis and RNA sequencing. **B** Protein level changes at relapse/diagnosis were determined for each patient ($N = 5$) and analyzed with a moderated 1-sample $t$-test. Average log2(fold change) of each protein is plotted against its –log10($p$-value). Top regulated proteins passing the 0.1 FDR significance cutoff are highlighted in color. **C** Western blot validation of top candidates in an independent patient cohort of primary patient samples obtained pre-treatment and at relapse ($N = 13$ patient samples). **D** Median normalized protein intensities (log2 TMT intensities) of CDK6, TRIP13, RRM1 and CRBN in all 10 samples were plotted against their respective normalized RNA expression levels (log2 TPM values). Samples from the same patient are connected. Source data are provided as a Source Data file.

related, replication, and chromosome maintenance signatures in relapse samples and downregulation of phosphorylation, ATP synthesis, N-glycan biosynthesis and unfolded protein response signatures. The gene signatures significantly enriched in the proteomic and phosphoproteomic data were not corroborated by the RNA sequencing data, indicating post-translational regulatory mechanisms.

**Protein expression of CDK6, TRIP13, and RRM1 is independent of CRBN.** As impaired CRBN-CRL4 E3 ligase activity due to mutation, deletion or downregulation leads to altered IMiD-sensitivity in multiple myeloma cell lines and patients[9–11], we examined its status in the five patients included in our proteomic analyses. No difference in CRBN RNA, protein or phosphorylation

levels were observed between pre-treatment and relapsed patient samples (Fig. 1D). Four out of five patient samples were analyzed also by exome sequencing and none was found to harbor mutations in members of the CRBN-CRL4 E3 ligase complex[17]. In our independent patient sample cohort, we did not observe any correlation between CRBN and CDK6, TRIP13, or RRM1 protein levels (Supplementary Fig. 2B). Consistently, CRISPR/Cas9-mediated knockout of *CRBN* in myeloma cell lines did not alter expression levels of these proteins (Supplementary Fig. 2C). Furthermore, no association was observed between genetic alterations and CDK6, TRIP13, or RRM1 protein expression in patient samples.

**CDK6 protein is upregulated in in vitro generated, lenalidomide-resistant multiple myeloma cells.** In order to mimic IMiD

**Table 1 Top upregulated proteins detected in global proteomics with their RNAseq profile.**

| Gene | uniprot | log2(FC.protein) | FDR(protein) | log2(FC.RNA) | FDR(RNA) |
|------|---------|------------------|--------------|--------------|----------|
| TRIP13 | Q15645 | 2.78 | 0.03 | 1.56 | 0.15 |
| RRM1 | P23921 | 2.72 | 0.08 | 0.59 | 0.77 |
| NCAPD2 | Q15021 | 2.59 | 0.07 | 0.39 | 0.83 |
| NCAPH | Q15003 | 2.53 | 0.09 | 0.97 | 0.73 |
| MORF4L1 | Q9UBU8 | 2.21 | 0.06 | 0.24 | 0.74 |
| CDK6 | Q00534 | 2.08 | 0.06 | 0.30 | 0.69 |
| BAZ1B | Q9UIG0-2 | 1.99 | 0.07 | −0.07 | 0.94 |
| CHTF18 | Q8WVB6 | 1.88 | 0.07 | 1.75 | 0.15 |
| HNRNPU | Q00839-2 | 1.87 | 0.06 | 0.32 | 0.63 |
| KEAP1 | Q14145 | 1.46 | 0.06 | 0.43 | 0.71 |

Proteins passing the 0.1 FDR significance cutoff were ranked by their fold change and FDR. The corresponding fold changes and FDR from RNAseq are displayed.

**Table 2 Top downregulated proteins detected in global proteomics with their RNAseq profile.**

| Gene | uniprot | log2(FC.protein) | FDR(protein) | log2(FC.RNA) | FDR(RNA) |
|------|---------|------------------|--------------|--------------|----------|
| UPRT | Q96BW1 | −2.73 | 0.03 | −0.67 | 0.45 |
| DNAJC1 | Q96KC8 | −2.26 | 0.06 | −0.82 | 0.43 |
| FCRL2 | Q96LA5 | −2.24 | 0.06 | −1.33 | 0.18 |
| AUH | Q13825 | −2.23 | 0.06 | 0.16 | 0.90 |
| HYI | Q5T013 | −2.16 | 0.06 | −0.98 | 0.55 |
| HID1 | Q8IV36-2 | −2.14 | 0.06 | −1.29 | 0.24 |
| GSTP1 | P09211 | −1.81 | 0.06 | −0.29 | 0.76 |
| GLO1 | Q04760-2 | −1.74 | 0.06 | −0.01 | 1.00 |
| CYP20A1 | Q6UW02 | −1.70 | 0.06 | −0.96 | 0.52 |
| USO1 | O60763 | −1.63 | 0.06 | −0.77 | 0.31 |

Proteins passing the 0.1 FDR significance cutoff were ranked by their fold change and FDR. The corresponding fold changes and FDR from RNAseq are displayed.

resistance in vitro, we cultured MM.1S and LP-1 cells in the presence of different concentrations of lenalidomide. Cells cultured in the presence of 100 nM lenalidomide for several weeks had enhanced levels of CDK6 protein and were partially resistant to lenalidomide, highly consistent with the findings in lenalidomide-treated myeloma patients (Supplementary Fig. 4A).

In contrast, short-term treatment with lenalidomide or proteasome inhibitors for up to 72 h had no effect or even decreased protein levels of CDK6, showing that their expression is not directly induced by the drugs (Supplementary Fig. 4B, C).

**Overexpression of CDK6 impairs IMiD sensitivity**. To investigate whether upregulation of CDK6, TRIP13, and RRM1 protein levels are causally linked to drug resistance, we induced their expression in MM.1S and OPM2 cell lines using lenti- and retroviral expression vectors (Fig. 2A, C, Supplementary Fig. 5A). TRIP13 overexpression did not alter drug sensitivity in MM.1S cells (Supplementary Fig. 5B–F). In contrast, both CDK6 and RRM1 overexpression in MM.1S and OPM2 cells reduced sensitivity to lenalidomide as well as to pomalidomide (Fig. 2B, D, Supplementary Fig. 5B, C). The effect of CDK6 was kinase-dependent since the introduction of a kinase-dead mutant CDK6 K43M[35] was not able to rescue cells from lenalidomide exposure (Fig. 2D). High CDK6 expression levels slightly enhanced sensitivity towards melphalan and dexamethasone in one cell line each (Supplementary Fig. 6). No effect was observable for bortezomib. In aggregate, these data imply that high CDK6 and RRM1 expression selectively reduce IMiD sensitivity in multiple myeloma cell lines.

**CDK6 kinase inhibition sensitizes multiple myeloma cells to IMiDs**. Given that CDK6 upregulation was found in lenalidomide-resistant patients and induced expression reduced lenalidomide

sensitivity, we next tested the effects of the CDK6 inhibitor palbociclib[36,37] in multiple myeloma cell lines. Palbociclib had no to moderate activity in multiple myeloma cell lines with 5 out of 10 responding (Fig. 3A, C, Supplementary Fig. 7A). However, palbociclib markedly enhanced the anti-multiple myeloma effects of IMiDs when both drugs were combined with high synergy scores (Fig. 3A, B, Supplementary Figs. 7A, 8A, B, E). This effect was observed across all cell lines expressing CDK6 at various levels (Supplementary Fig. 7B). Remarkably, this included multiple myeloma cell lines that are naturally insensitive to IMiDs like L363 (Fig. 3C, D) and AMO-1, and synergy was observed at low drug concentrations corresponding to plasma levels in treated patients[38]. In acquired lenalidomide-resistant cells with increased CDK6 protein levels, as well as in CDK6 overexpressing cells, the addition of palbociclib restored IMiD-sensitivity to levels similar as in parental cells (Fig. 3E, F). These data show that palbociclib treatment increases the sensitivity to lenalidomide and pomalidomide in multiple myeloma cells. In contrast, combined treatment of palbociclib and melphalan, bortezomib or dexamethasone was mostly additive (Supplementary Fig. 8C, D).

**Bifunctional PROTACs degrading CDK6 and IKZF1/3 possess intramolecular synergy**. We next investigated an alternative way to inactivate CDK6 using protein degradation. Proteolysis targeting chimeras (PROTACs) are bifunctional molecules which comprise two linker-connected moieties that simultaneously bind a target protein and an E3 ubiquitin ligase[39]. Like IMiDs, PROTACs hijack E3 ubiquitin ligases and induce ubiquitination and degradation of the target protein. We and others have recently described PROTACs that effectively target CDK6 for proteasomal degradation through hijacking the CRBN- or von Hippel-Lindau (VHL) E3 ligase (Fig. 4A, B)[40,41]. We tested the anti-proliferative effects of the CDK6-selective, CRBN-recruiting PROTAC BSJ-03-

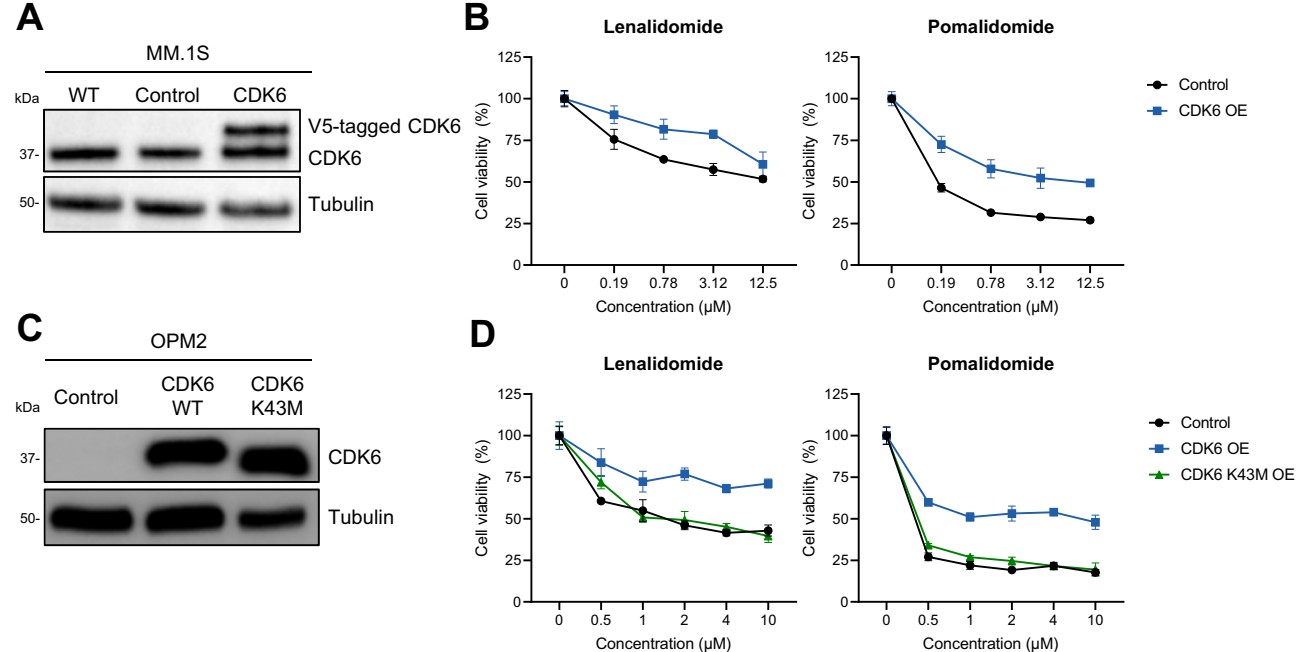

**Fig. 2 High expression levels of CDK6 confers augmented IMiD-resistance. A** Overexpression of CDK6 in MM.1S cells using lentiviral transduction confirmed through western blot analysis. **B** Cell viability of CDK6-overexpressing MM.1S cells upon 96 h treatment with lenalidomide and pomalidomide at indicated concentrations. ($N = 3$ biologically independent replicates). **C** Overexpression of CDK6 in OPM2 cells using retroviral transduction confirmed through western blot analysis. **D** Cell viability of CDK6 WT or K43M-overexpressing OPM2 cells upon 96 h treatment with lenalidomide and pomalidomide at indicated concentrations. ($N = 3$ biologically independent replicates) Control denotes empty vector. Cell viability is normalized to respective DMSO conditions. Data represent the mean ± SD of biological triplicates. Source data are provided as a Source Data file.

123 or the VHL-recruiting PROTAC CST528 that degrades both CDK4 and CDK6. Consistent with the results for the kinase inhibitor palbociclib, PROTAC-mediated CDK6 degradation reduced multiple myeloma cell growth in a subset of cell lines. Combination of IMiDs with the VHL-recruiting PROTAC CST528 had synergistic effects with IMiDs consistent with the results observed for the CDK6 kinase inhibitor palbociclib (Fig. 4C, D)[41]. In contrast, combination of IMiDs with a CRBN-hijacking, CDK6-specific PROTAC (BSJ-03-123) showed antagonistic effects that are likely due to the competition for the CRBN E3-ligase (Supplementary Fig. 9).

We next tested a CRBN-hijacking, pomalidomide-based PROTAC, YKL-06-102, that retains the activity of pomalidomide and potently induces both degradation of CDK6 and IMiD neosubstrates IKZF1 and IKZF3[40]. YKL-06-102 significantly reduced viability in all ten multiple myeloma cell lines tested, including those with a low IMiD sensitivity (Fig. 4E and F, Supplementary Fig. 10). These results show that CRBN-hijacking PROTACs targeting CDK6, IKZF1, and IKZF3 simultaneously are highly effective in multiple myeloma cells through intramolecular synergy.

**Combination treatment of pomalidomide and palbociclib is highly effective in vivo.** To test whether the combination of IMiDs with CDK6 inhibition has therapeutic efficacy in vivo, we conducted a study in the MM.1S xenograft model (Fig. 5A). MM.1S cells were injected subcutaneously and mice were randomized for treatment groups after 19 days when tumors reached 200 mm³. Treatment was performed for 17 days, with pomalidomide and palbociclib being orally administered at 5 and 50 mg/kg, respectively. Pomalidomide and palbociclib as monotherapy significantly delayed tumor growth, while combination therapy reduced tumor volumes below detection limits after 2 weeks (Fig. 5B, Supplement Fig. 11). After cessation of treatment, tumor

growth resumed, indicating that prolonged treatment with both drugs is necessary to prevent multiple myeloma relapse. The potent suppression of tumor growth in the combination group translated to a significant improvement in surivival as compared to mice that recieved pomalidomide or palbociclib alone (Fig. 5C).

Intraperitoneal application of the CDK6/IKZF1/IKZF3-degrading PROTAC YKL-06-102 at a maximal applicable dosage of 5 mg/kg per day (due to low solubility) significantly delayed tumor growth as compared to control treated mice, yet the effect was not better than pomalidomide or palbociclib alone, likely due to lower bioavailability in vivo (Supplementary Fig. 11).

**The synergistic effect of IMiDs and CDK6 inhibition is independent of RB1 and cell cycle progression.** RB1 is one of the major substrates of CDK6 and palbociclib treatment resulted in reduced phosphorylation of RB1 and G1 cell cycle arrest in MM cell lines, consistent with previous studies in cancer[42,43]. Heterozygous chromosome 13q/*RB1* deletions are among the most frequent genetic alterations in multiple myeloma and complete loss is observed in heavily treated patients, implying that it contributes to drug resistance[14]. We therefore tested whether *RB1* knockout through CRISPR/Cas9 affects sensitivity to pomalidomide, palbociclib or the combination treatment with CDK6 inhibition or degradation in MM.1S cell line. In line with studies in breast cancer, *RB1* knockout reduced sensitivity to palbociclib (Supplementary Fig. 12). However, the synergistic effects of CDK6 inhibition and IMiDs were retained in *RB1* knockout cells, demonstrating that the sensitization to IMiDs is independent of functional RB1.

**CDK6 inhibition reverses a relapse-associated protein signature.** To investigate the effects of CDK6 inhibition and the basis for the synergy with IMiDs in multiple myeloma, we

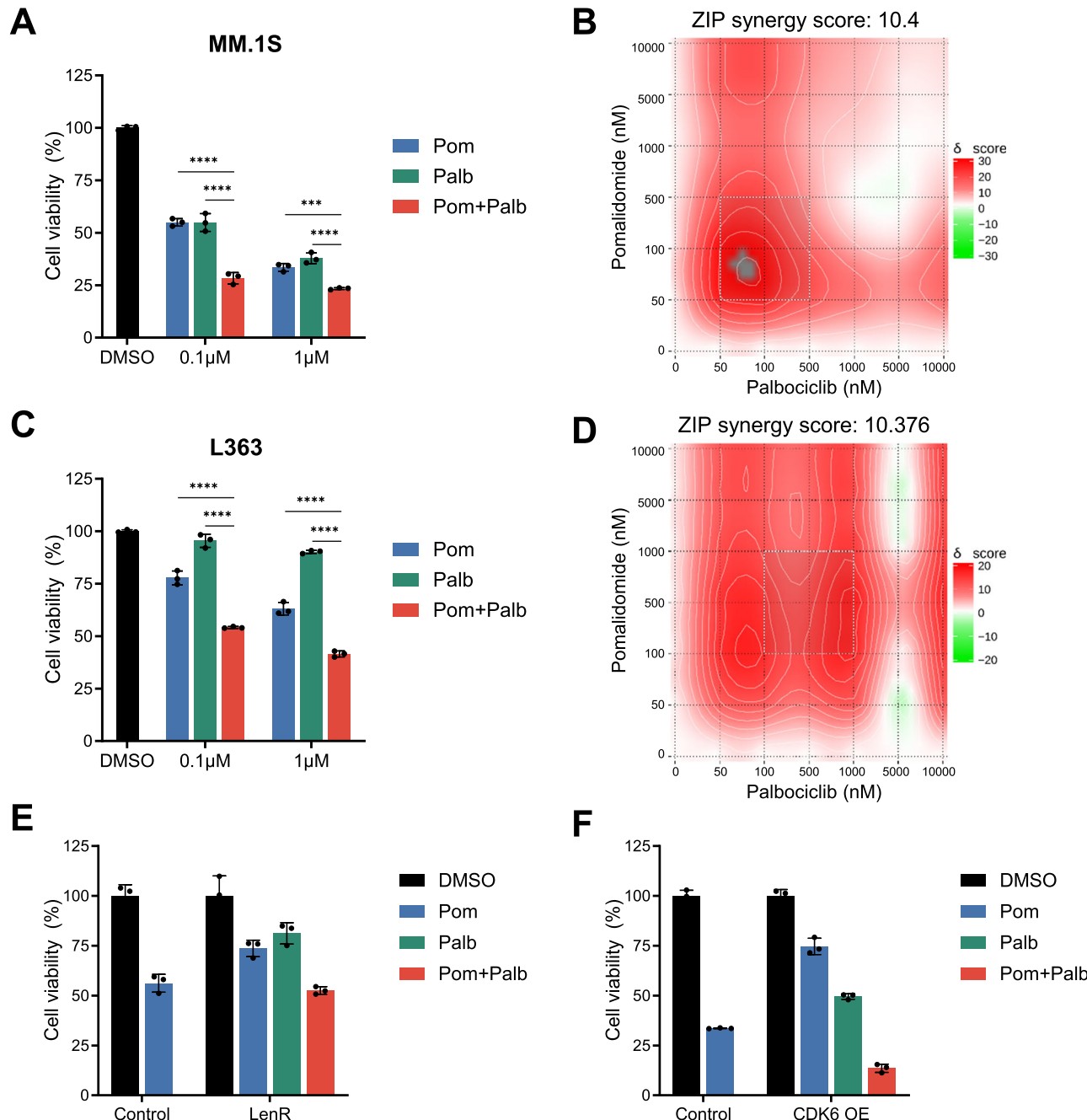

**Fig. 3 CDK6 inhibition by palbociclib synergizes with IMiD treatment of multiple myeloma cells. A** Cell viability of MM.1S cells treated 96 h with pomalidomide (pom), palbociclib (palb), or in combination. ($N = 3$ biologically independent replicates). **B** Synergy map of MM.1S cells treated with palbociclib in combination with lenalidomide or pomalidomide. **C** Cell viability of L363 cells treated with pomalidomide, palbociclib, or in combination ($N = 3$ biologically independent replicates). **D** Synergy map of L363 cells treated with palbociclib in combination with lenalidomide or pomalidomide. **E** Cell viability of MM.1S LenR cells ($N = 3$ biologically independent replicates) and **F** OPM2 CDK6 OE cells upon combination treatment of pomalidomide and palbociclib ($N = 3$ biologically independent replicates) Synergy levels are indicated with ZIP synergy scores. Synergy maps were generated with SynergyFinder[75]. Cell viability is normalized to respective DMSO conditions. Data represent the mean ± SD of biological triplicates. One-way ANOVA is applied. $P$ values are displayed as follows: n.s. = $P > 0.05$; *$P \leq 0.05$; **$P \leq 0.01$; ***$P \leq 0.001$; ****$P \leq 0.0001$. Source data are provided as a Source Data file.

performed quantitative proteomic and phosphoproteomic analyses in MM.1S cells treated with pomalidomide, palbociclib, and CDK6-specific PROTACs alone or in combination (Fig. 6A, Supplementary Data 3). Global proteome analysis of drug-treated MM.1S cells identified 7460 proteins, of which 240 and 179 were significantly regulated by treatment with 24 h pomalidomide or palbociclib, respectively (0.1 FDR). In line with previous results,

CRBN neo-substrates IKZF1, IKZF3, and ZFP91 were the most downregulated proteins in pomalidomide treated samples. The effect of palbociclib and the CDK6-specific PROTAC BSJ-03-123 on protein levels highly correlated (Supplementary Fig. 13A, B).

When we compared the significantly regulated proteins in patient samples and drug-treated MM.1S cells, we observed a striking pattern where proteins upregulated in patient samples at

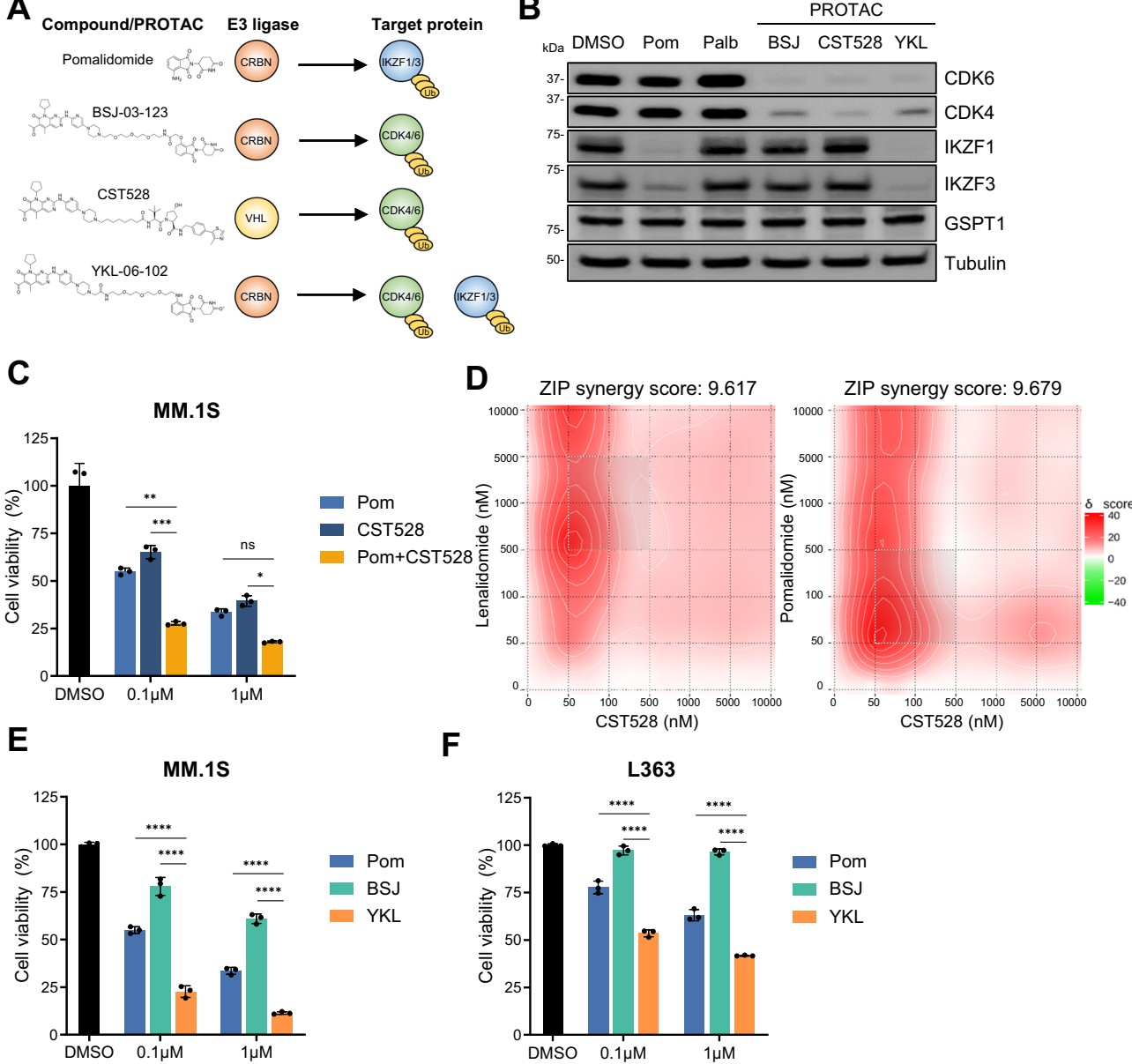

**Fig. 4 CDK6 inhibition synergizes with IKZF1/3 degradation. A** Chemical structure, hijacked E3 ligase, and targets of pomalidomide (pom), PROTAC BSJ-03-123 (BSJ), PROTAC CST528, and PROTAC YKL-06-102 (YKL). **B** Western blot analysis of MM.1S cells treated with 1 μM of pomalidomide, palbociclib, BSJ-03-123, CST528, and YKL-06-102 for 16 h ($N = 3$ biologically independent replicates). **C** Cell viability of MM.1S cells treated 96 h with pomalidomide, CST528 and in combination ($N = 3$ biologically independent replicates). **D** Synergy map of MM.1S cells treated with CST528 in combination with lenalidomide or pomalidomide. **E** Cell viability of 96 h treatment with pomalidomide, BSJ-03-123 and YKL-06-102 in MM.1S cells ($N = 3$ biologically independent replicates) and **F** L363 cells ($N = 3$ biologically independent replicates). Synergy levels are indicated with ZIP synergy scores. Synergy maps were generated with SynergyFinder[75]. Cell viability is normalized to respective DMSO conditions. Data represent the mean ± SD of biological triplicates. One-way ANOVA is applied. *P* values are displayed as follows: n.s. = $P > 0.05$; *$P \leq 0.05$; **$P \leq 0.01$; ***$P \leq 0.001$; ****$P \leq 0.0001$. Source data are provided as a Source Data file.

relapse were downregulated by CDK6 inhibition or degradation in MM.1S cells and vice versa (Fig. 6B, Supplementary Data 4). In detail, we found an overlap of 37 proteins upregulated in the relapse samples and downregulated by CDK6 inhibition (cluster a), as well as 54 proteins significantly downregulated in relapse and upregulated by CDK6 inhibition (cluster c) (Supplementary Fig. 13C). Among the CDK6 regulated proteins included in the signature were the two top upregulated proteins in relapse, TRIP13 and RRM1[30,31]. Other proteins upregulated in relapse and targeted by CDK6 inhibition are comprised of cell cycle-related genes (CDK2, MCM3/5, NCAPH, NCAPD2, PDCD2L,

USP16), DNA damage regulators (PAXIP1, RAD18, and MLH1) and transcriptional/ epigenetic regulators (DNMT1, BTF3, EDRF1, KEAP1, MORF4L1). Upregulation of the genome integrity safeguard PAXIP1 links CDK6 to a factor that has been identified to be highly selective for myeloma cell survival in the Dependency Map studies[44] and the higher abundance of the oxidative stress sensing E3 ligase KEAP1 may indicate a connection to oxidative metabolism. The downregulated proteins in cluster c are dominated by 27 mitochondrial genes including key factors for leucine catabolic metabolism (AUH), TCA cycle (IDH3B), the electron transport chain (NDUFA10, COX5B),

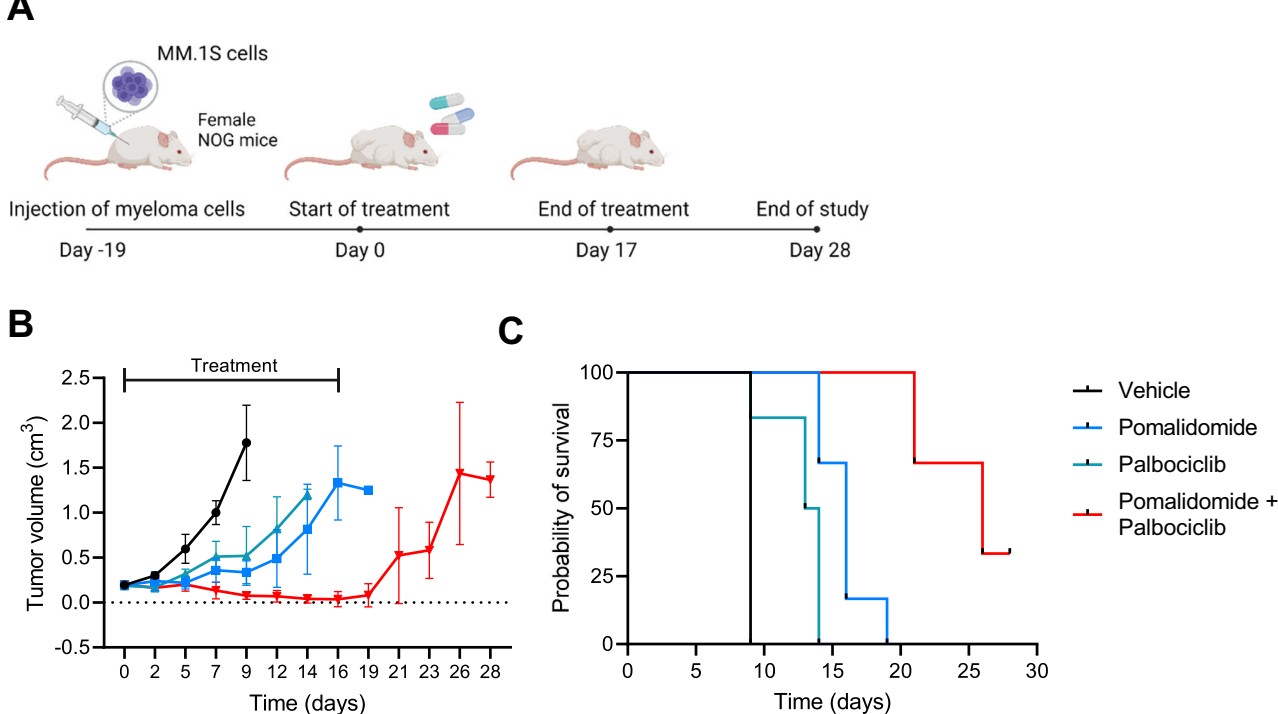

**Fig. 5 Combination treatment of pomalidomide and palbociclib has high therapeutic efficacy in multiple myeloma in vivo. A** MM.1S cells were injected into NOG mice and treatment started 19 days after injection of myeloma cells when tumors were ~0.2 cm³. Mice were treated on a daily basis with vehicle control, pomalidomide (5 mg/kg), palbociclib (50 mg/kg) or the combination p.o. for 17 days and observed until day 28. **B** Tumor growth in treated mice of monotherapy of pomalidomide and palbociclib, and in combination. Mice were taken out of the study when tumors exceeded a size of 1.2 cm³. **C** Survival of the four groups. Statistical differences were analyzed by log-rank Mantel–Cox test. All comparisons of survival curves resulted in $P$-vaues < 0.01. Data represent mean ± SD of biological replicates. Group size: $n = 5$ for vehicle group; $n = 6$ for pomalidomide treatment group; $n = 6$ for palbociclib treatment group; $n = 6$ for pomalidomide + palbociclib treatment group. Figure **A** was created with BioRender.com. Source data are provided as a Source Data file.

ATP synthase (ATP5D, ATP5J), and fatty acid metabolism (ACAD9, ECHS1, HADHB, MLYCD, MCEE). Similar to the effects observed here it has been reported that the signling axis CDK4-RB1-E2F represses mitochondrial oxidative metabolism[45].

We also observed lower levels of IKZF1/3 and MYC in the combined treatments as compared to single treatments alone. (Fig. 6C, Supplementary Fig. 14A, B). The enhanced inactivation of essential multiple myeloma transcription factors likely contributes to the synergistic effects of IMiDs and CDK6 inhibition.

**Identification of CDK6 substrates by phosphoproteomics in multiple myeloma.** To identify CDK6 substrates in multiple myeloma, we performed phosphoproteomics analyses in MM.1S cells treated with palbociclib or the CDK6-specific PROTAC BSJ-03-123. We identified 29,811 phosphopeptides across all samples of which 122 phosphopeptides on 96 proteins were significantly downregulated after 3 h of palbociclib or BSJ-03-123 treatment (0.1 FDR). In addition, phosphorylation status of 558 proteins (1022 phosphopeptides) not regulated on global protein level was specifically downregulated after 24 h palbociclib treatment, with an overlap of 45 proteins to the 3 h timepoint, which we deemed particularly interesting, since these phosphosites were down-regulated at an early time point and in a sustained manner (Supplementary Data 3, Supplementary Fig. 15A). The effect of palbociclib and the CDK6-specific PROTAC BSJ-03-123 on phosphopeptide levels was highly correlated (Supplementary Fig. 15B). Identified phosphorylation targets comprised the known CDK4/6 substrates RB1, RBL1, RBL2, and CDKN1A as well as several proteins that were deregulated in relapse samples (DNTM1, GMPS, KLHDC4, NCAPD2, NCAPH, GLYR1,

NOP56), providing a functional link between CDK6 kinase function and CDK6-regulated protein levels. In addition, lower phosphorylation levels were observed in palbociclib treated cells in several transcriptional regulators including RNF169 and ZBTB38, as well as known CDK6 interactors and substrates JUN and FOXM1 (Supplementary Data 3)[46,47]. However, for many proteins deregulated on the protein level, including RRM1 and TRIP13, we did not detect lower phosphorylation levels after CDK6 inhibition, indicating that these proteins are not direct CDK6 substrates. RT-qPCR analyses revealed that TRIP13, RRM1 and other proteins tested had reduced mRNA levels after CDK6 inhibition or inactivation, implying CDK6-dependent transcriptional regulation as has been previously reported (Fig. 6D)[46].

**Discussion**

Drug resistance is one of the biggest challenges in cancer therapy. Genetic alterations revealed in comprehensive sequencing studies do not sufficiently explain the emergence of resistance in most cases, implying non-genetic mechanisms[48]. Here we applied an integrated proteomics and transcriptomics approach in primary multiple myeloma matched pre-treatment/resistant samples that identified and validated a targetable CDK6 governed protein resistance signature. Only a few proteomic studies of multiple myeloma have been previously published[26,49,50]. Our dataset containing over 6000 proteins and 20,000 phosphopeptides represents a deep proteomic study of longitudinal samples from relapsed multiple myeloma patients. It provides proof of principle for feasibility and clinical relevance of in-depth proteomic analysis in multiple myeloma. We observed a low correlation of protein and RNA levels implying a high degree of post-

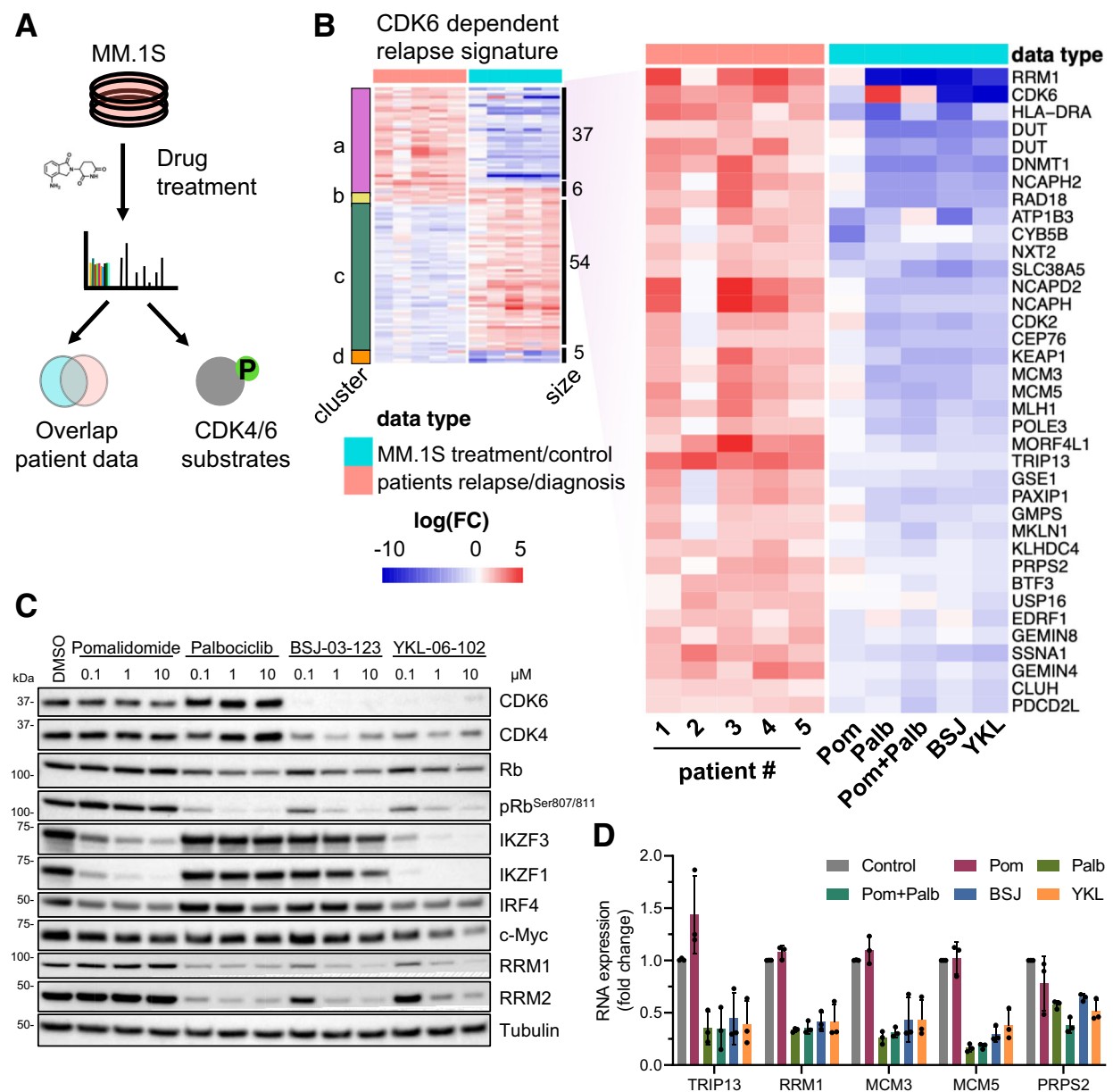

**Fig. 6 Targeting CDK6 in myeloma cells reverses a relapse protein signature. A** Proteomic and phosphoproteomic changes of MM1.S cells treated with different drugs and combinations were assessed with TMT-based proteomics. **B** Combined heatmap displaying protein levels of paired-patient data and cell perturbation data. MM.1S cells were treated for 24 h with pomalidomide (pom), palbociclib (palb), in combination (pom + palb), BSJ-03-123 (BSJ), and YKL-06-102 (YKL). **C** Western blot analysis of MM.1S cells treated for 16 h with respective treatments at indicated concentrations (N = 3 biologically independent replicates). **D** mRNA levels of downstream CDK6 targets in MM.1S cells upon treatment of 1 μM with pomalidomide, palbociclib, or in combination, BSJ-03-123 and YKL-06-102 (N = 3 biologically independent replicates) mRNA expression is normalized to respective GAPDH levels and DMSO conditions. Data represent the mean ± SD of biological triplicates. Source data are provided as a Source Data file.

transcriptional and post-translational regulation in multiple myeloma. Indeed, the most upregulated and functionally relevant proteins identified by our proteomics approach would have been missed by RNA sequencing. This is in line with previous studies in other tumors highlighting that protein level and activity cannot necessarily be inferred from RNA expression[21,51,52].

Unexpectedly, proteins known to be involved in pathways targeted by routinely used myeloma drugs like CRBN, IKZF1, proteasome subunits, or the glucocorticoid receptor were not found deregulated at relapse. However, several of the proteins we identified as significantly deregulated have been previously implicated in multiple myeloma. TRIP13, an AAA-ATPase involved in genome stability[53], is among the top-ranking genes included in the GEP70, a

highly validated high-risk RNA expression signature in multiple myeloma[18,29,30]. TRIP13 overexpression is a driver of B-cell malignancies[28] and is functionally linked to drug resistance in multiple myeloma and other cancers[30], and inhibition of TRIP13 is toxic to multiple myeloma cells[54]. RRM1, together with RRM2, forms the ribonucleotide reductase heterodimeric tetramer that is involved in DNA damage repair[55]. Multiple myeloma cells were shown to be dependent on RRM1 and high RRM1 RNA expression is associated with an adverse outcome[31]. In line with this, we found RRM1 overexpression to reduce IMiD sensitivity in multiple myeloma cells.

CDK6, together with CDK4, is a vital regulator of the cell cycle through regulation of RB1[42,56–58]. CDK6 has been found

upregulated in a wide range of cancer, including multiple myeloma[32,46,59]. We found that CDK6 overexpression reduces sensitivity to IMiDs, but not to other drugs in multiple myeloma. Blocking CDK6 through palbociclib or by protein degradation via PROTACs inhibited cell cycle progression and proliferation consistent with previous reports[32,36]. Palbociclib was able to overcome IMiD resistance and showed strong synergy with lenalidomide and pomalidomide in vitro and in vivo. CDK6-specific PROTACs hijacking CRBN were antagonistic with IMiDs due to E3 ligase competition. This issue could be overcome by either hijacking two different E3 ligases, CRBN and VHL, or a PROTAC, such as YKL-06-102, that simultaneously targets IKZF1, IKZF3, and CDK6, achieving intramolecular synergistic effects even in IMiD-insensitive multiple myeloma cells. However, the low activity of this PROTAC in the MM.1S xenograft model shows that the pharmacologic properties of PROTACs need to be optimized for application in vivo.

Our data imply that the effects of CDK6 in multiple myeloma depend on its kinase function but are independent of RB1, the CDK4/6 substrate which is most relevant for other cancers. Our proteomic analyses in multiple myeloma cell lines revealed that CDK6 governs many of the proteins deregulated at relapsed/resistant disease, either through direct phosphorylation of proteins or by indirect alterations of the transcriptional activity of genes including TRIP13 and RRM1, and is thus a master regulator of a relapse-associated program. The CDK6-dependent relapse signature contains proteins connected to DNA damage repair, cell cycle, and metabolic pathways including electron transport chain, energy transfer, and fatty acid metabolism. Antagonizing this signature by CDK6 inactivation in conjunction with IMiDs resulted in a significant and fast downregulation of the essential transcription factor MYC, further highlighting the synergistic effects of IKZF1/3 and CDK6 inactivation. While many of the relapse-associated proteins are regulated by CDK6, the mechanism that drives elevated CDK6 protein expression itself is not clear. We couldn't find any associations with genetic alterations and the lack of correlation with RNA expression implies post-transcriptional and/ or post-translational mechanisms. This, as well as the exact downstream functions of CDK6 in resistant multiple myeloma, needs to be evaluated in detail in future studies.

Several CDK4/6 inhibitors are approved by the FDA and EMA for advanced breast cancer and are being investigated in hematologic malignancies including acute lymphoblastic leukemia[60] and acute myeloid leukemia[33,61,62]. In multiple myeloma, a phase I/II clinical trial that combined palbociclib with bortezomib and dexamethasone showed that this regimen is feasible and has antitumor activity[63]. Here, we show that combining CDK6 inhibition with IMiDs is particularly synergistic in multiple myeloma. Although both drugs cause cytopenia as an unwanted side-effect, our data suggest that low dosages of palbociclib may be sufficient to sensitize multiple myeloma cells to IMiDs. PROTACs have recently entered clinical trials[64] and PROTACs with combined IKZF1/IKZF3/CDK6 activity provide an attractive alternative with single drug synergistic effects. CDK6 protein levels may provide a biomarker to identify patients who would benefit most from this therapeutic combination. In conclusion, our results identify CDK6 as a master regulator in treatment-resistant relapsed multiple myeloma and provide a strong rationale for further investigating CDK6 inhibition together with IMiDs in multiple myeloma.

## Methods

**Study cohort.** Five multiple myeloma patients with paired pre-treatment and relapsed samples were included in the proteomics study. Additional 13 samples from an independent cohort of newly diagnosed or relapsed/refractory multiple

myeloma patients were analzyed with western blotting. Patient characteristics and treatment are summarized in Supplementary Fig. 1. All samples were obtained from the iliac crest of patients and were CD138+ enriched by MACS (Miltenyi, Cologne, Germany). For patients #1, #3, #4, and #5 whole-exome sequencing data is reported in a previous study[17]. All patients provided written informed consent according to the Declaration of Helsinki and the study was approved by the institutional review board (IRB) of Ulm University.

**Cell culture.** Multiple myeloma cell lines MM.1S, OPM2, NCI-H929, L363, LP1, RPMI-8226, AMO-1, INA-6, JJN3, and KMS12BM were obtained from ATCC (Manassas, Virginia, USA) and the German Collection of Microorganisms and Cell Cultures GmbH (DSMZ, Braunschweig, Germany), and maintained in RPMI-1640 medium (Merck KGaA, Darmstadt, Germany) containing 10% fetal bovine serum (FBS) and supplemented with 1% penicillin/streptomycin and 1% L-glutamine. INA-6 cells were supplemented with IL-6. Cells were maintained at 37 °C with 5% $CO_2$ in humidified atmosphere.

**Treatment of MM1S for proteomic and phosphoproteomic analysis.** MM.1S WT cells were subjected to respective treatments for either 3 or 24 h in biological duplicates. Pomalidomide treatment was administered at a final concentration of 10 μM, while palbociclib, BSJ-03-123, and YKL-06-102 were administered at 1 μM. Control cells were treated with solvent only. Cells were pelleted and washed three times with cold 1× PBS.

**Proteomics and phosphoproteomics sample preparation.** Samples for global proteome and phospho-proteome profiling were analyzed with isobaric TMT[21]. Samples were lysed at 4 °C with urea lysis buffer (8 M urea, 50 mM Tris (pH 8), 150 mM NaCl) supplemented with protease inhibitors (2 μg/ml aprotinin, 10 μg/ml leupeptin 1 mM phenylmethylsulfonylflourid) and phosphatase inhibitors (10 mM NaF, phosphatase inihibitor cocktail 1 and 2, Sigma Aldrich). Extracted proteins were reduced with 5 mM dithiothreitol for 1 h and alkylated with 10 mM iodoacetamide for 45 min in the dark. Sequencing grade LysC (Wako) was added at a weight to weight ratio of 1:50. After 2 h, samples were diluted 1:4 with 50 mM Tris–HCl pH 8 and sequencing grade trypsin (Promega) was added at a weight to weight ratio of 1:50. Digestion was completed overnight and subsequently acidified samples were desalted with Sep-Pak C18 cc Cartridges (Waters). Dried samples were resuspended in 10 mM HEPES (pH 8.5) and peptide concentration was determined. For relative quantification of proteins in multiple myeloma patient samples, 10 μg peptides for each sample were labeled with 10-plexing TMT (TMT-10; Thermo Fisher Scientific) and combined into one TMT plex. For quantification of phosphopeptides in multiple myeloma patient samples, 150 μg of peptides for the samples 1–9 and 23 μg of starting material for sample 10 were labeled with TMT-10 reagents and combined into a second plex. TMT-channel annotation of patient proteomic data is available in Supplementary Data 1. For global and phosphoproteome analysis of MM.1S drug-treated cells, 200 μg peptides of each sample were labeled with 11-plexing TMT (TMT-11; Thermo Fisher Scientific) and the samples were randomly distributed into two TMT plexes. For the MM.1S experiment, an internal reference sample composed of equal amounts of peptide material from all samples was included in each TMT 11-plex to provide a standard for relative quantification. TMT-channel annotation of MM.1S proteomic data is available in Supplementary Data 3. Combined TMT samples were fractionated offline into 24 fractions using high-pH reversed phase chromatography. For analysis of the global proteome, 500 ng peptides of each fraction were analyzed with LC–MS/MS. The remaining material was pooled into 12 fractions for each plex and subjected to phosphopeptide enrichment with immobilized metal affinity chromatography (IMAC) automated on an AssayMap Bravo System (Agilent)[65].

**Liquid chromatography mass spectrometry.** Mass spectrometry raw data was acquired on a Q-Exactive HF-X connected to an EASY-nLC 1200 system (both Thermo Fisher Scientific). Each HpH fraction was measured individually as a single LC-MS injection. Samples were separated online on a 25 cm column packed inhouse with C18-AQ 1.9 μm beads (Dr. Maisch Reprosil-Pur 120). A gradient of mobile phase A (0.1% formic acid and 3% acetonitrile in water) and mobile phase B (0.1% formic acid, 90% acetonitrile in water) was used to separate the samples online at a flow rate of 250 μl/min. Mobile phase B was ramped from 4% to 30% in the first 88 min, followed by an increase to 60% B in 10 min and a plateau of 90% B for 5 min. Temperature of the column was kept constant at 45 °C. MS data was acquired in data-dependent acquisition and profile centroid mode. MS1 scans were acquired at 60,000 resolution, scan range of 350–1500 $m/z$, automatic gain control (AGC) target value of 3e6 and maximum injection time (IT) of 10 ms. The 20 most abundant ions were picked for fragmentation with normalized collision energy (NCE) set to 32 and 0.7$m/z$ isolation window. MS2 scans were acquired at 45,000 resolution, fixed first mass 120 $m/z$, AGC target value of 3e5 and maximum IT of 86 ms. Ions with charge state 1, 6 or higher were excluded from fragmentation and dynamic exclusion was set to 30 s. LC–MS parameters for phosphoproteomic analysis was the same with the exception of MS2 maximum IT that was set to 120 ms.

**Mass spectrometry raw data searching with MaxQuant**. Raw data was analyzed with MaxQuant (Version 1.6.3.3)[66] and searched against the human reference proteome (UP000005640) downloaded from UniProt in 01/2017 (https://ftp.uniprot.org/pub/databases/uniprot/previous_releases/) and default protein contaminants included in MaxQuant. PIF filter was set to 0.5. Fixed modifications were set to carbamidomethylation of C. Variable modifications were set to M-oxidation and acetylation of protein N-termini including neo protein N-terms after cleavage of first methionine. A maximum of 5 modification per peptide were allowed. N-terminal acetylation and M-oxidation were used in protein quantification (unmodified counterpart discarded). Unique and razor peptides were used for quantification. TMT correction factors supplied by the manufacturer were applied.

**Mass spectrometry data analysis**. Analysis of MaxQuant output was performed with R(4.0.3) and R studio (Version 1.3.1093) statistical software environment. Statistical analysis in R was aided by the ProTIGY application provided by the Broad institute on GitHub (https://github.com/broadinstitute/protigy)[67]. Protein group files were filtered for reverse hits, potential contaminants and proteins only identified by site. Resulting proteins were filtered for proteins identified with at least two peptides and at least one unique peptide. Phosphoproteomic data was analyzed in a separate MaxQuant run and filtered for valid values across all samples. Due to limited protein amounts of the patient 5 diagnosis sample, patient 5 was excluded from the analysis of the phosphoproteomic data. For the analysis of multiple myeloma patient samples, log2 ratios (relapse/diagnosis) of proteins and phosphopeptides were calculated for each patient and normalized with median-MAD normalization. The processed data was analyzed with a two-sided moderated one-sample t-test (limma package)[68]. Prior to applying multiple testing correction with the Benjamini–Hochberg method, data was subjected to reproducibility filtering to remove outliers (global proteome 99% confidence interval; phosphoproteome 95% confidence interval with linear mixed effect models). As expected, the phosphoproteome showed higher variability compared to the global proteome, which is more stable towards dynamic modifications. We therefore applied a stronger outlier filter and more relaxed cutoff for the phosphoproteomic data. A cutoff of 0.1 and 0.12 FDR was implemented for global proteome (Fig. 1B) and phosphoproteome (Supplementary Fig. 3C) analysis of patients samples respectively. For the analysis of MM.1S drug treated cells, the log2 ratios (sample/internal standard) were calculated and normalized with median-MAD normalization. The resulting ratios were subsequently normalized to the control sample and analyzed with a two-sided moderated one-sample t-test. Resulting p-values were corrected for multiple testing with the Benjamini–Hochberg method. MM.1S phosphoproteomic data was subjected to reproducibility filtering (99% confidence interval) with Bland–Altman filtering to remove outliers prior to p-value correction. For MM1.S derived data, a 0.1 FDR significance cutoff was applied. The Phosphosite plus (PSP) database was used for identifying known CDK6 targets[69]. The processed proteomics data as well as TMT channel annotation is available in supplementary data files. The mass spectrometry proteomics data and search results have been deposited to the ProteomeXchange Consortium via the PRIDE partner repository[70] with the dataset identifier PXD021265.

**RNA-sequencing**. Library preparation was performed from 100 ng of input total RNA using the TruSeq Stranded Exome RNA Kit (Illumina, San Diego, CA, USA) according to the manufacturer's instructions. The pooled RNA libraries were sequenced on an Illumina HiSeq2000 with 50 bp single-end reads with an average coverage of $36.6 \times 10^6$ reads per sample. RNA-Seq data was aligned and quantified with STAR[71] and mRNA reads were identified using an in-house analysis pipeline detecting exons in a shuffled order. To increase comparability with proteomics data, TPM RNAseq data was processed like TMT-proteomics data and median-MAD normalized log2 ratios (relapse/diagnosis) were analyzed with a two-sided moderated one-sample t-test. RNA transcripts were matched with the protein groups file and Pearson correlation of individual genes to proteins was calculated across all 10 samples (median normalized values). Processed RNAseq data is available in the supplements. Gene set enrichment analysis was performed using GSEA and the Molecular Signatures Database (MsigDB)[72]. RNA-sequencing data is available on Gene Expression Omnibus (accession number GSE162403).

**Fluorescence in situ hybridization (FISH)**. Fluorescence in situ hybridization was performed on plasma cells isolated from patients. Genetic regions of interest specific for the diagnosis for multiple myeloma and their translocation partners were detected. FISH was performed according to standardized protocols using commercially available probes (Abbott Laboratories, Chicago, United States; MetaSystems, Altlussheim, Germany)[73].

**In vivo experiment**. $1 \times 10^7$ MM.1S cells were injected subcutaneously into 6–8-week-old female NOG mice (strain: NOD.Cg-Prkdc$^{scid}$ Il2rg$^{tm1Sug}$/JicTac). Mice for the study were obtained from Taconic (Leverkusen, Germany). The in vivo experiments were performed at EPO GmbH Berlin, Germany. The animals were group-housed (max. 5 mice/cage) in individually ventilated cages (IVC type GM 500, Techniplast) at temperature of $22 \pm 2$ °C, humidity of $50 \pm 10$%, at 12 h dark–light cycles.

Tumor-bearing mice were randomized after 19 days when tumors reached 200 mm$^3$ and treatments were started. Pomalidomide was administered at 5 mg/kg

while palbociclib was administered at 50 mg/kg. Both drugs were administered daily via oral gavage. YKL-06-102 was administered daily at 5 mg/kg via intraperitoneal injections. Treatment was performed for 17 days and thereafter, remaining animals were observed for another 11 days. Mice were euthanized when tumor volume reached 1200 mm$^3$. The animal study was conducted in compliance with the United Kingdom Coordinated Committee on Cancer Research guidelines and have been approved and authorized by the Landesamt für Gesundheit und Soziales, Berlin, Germany (approval No. G 0333/18).

**Generation of lenalidomide-resistant cell lines**. MM.1S and LP-1 myeloma cells were cultured for over 3 weeks in the continuous presence of lenalidomide at 10, 100 and 1000 nM. Lenalidomide-resistance was confirmed with drug sensitivity assay with lenalidomide treatment.

**Plasmids and viral transduction**. The following plasmids were used: pLenti.6.2.V5-DEST, pRSF91-GFP-T2A-Puro, pLKO5d.SSF.SpCas9.P2a.BSD and pLKO5.hU6.sgRNA.dTom[61,74]. Complementary DNA (cDNA) of CDK6 and TRIP13 were cloned into pRSF91-GFP-T2A-Puro plasmid. pLenti.6.2.V5-DEST containing CDK6 WT and CDK6 K43M mutant were kind gifts from Claudia Scholl (National Center for Tumor Diseases and German Cancer Research Center (DKFZ)). sgRNAs targeting RB1 were cloned into pLKO5.hU6.sgRNA.dTom.

For generation of lentiviral and retroviral vectors, HEK293T cells were transfected with constructs along with their respective packaging and envelope vectors. Viral supernatants were harvested 48 h after transfection and were used to transduce human multiple myeloma cell lines.

**Generation overexpressing and knockout cells**. Human multiple myeloma cell lines were transduced with virus containing with or without the CDK6 construct in backbones pLenti.6.2.V5-DEST or pRSF91-GFP-T2A-Puro. Transduced cells were selected with blasticidin and puromycin (InvivoGen, San Diego, USA) respectively.

Human multiple myeloma cell lines were transduced with virus containing pLKO5d.SSF.SpCas9.P2a.BSD and cells were selected with blasticidin (Invivogen, San Diego, USA). Cells were then transduced with respective CRBN and RB1 sgRNA constructs cloned into pLKO5.hU6.sgRNA.dTom, along with luciferase and POLR2A controls. Transduction success was confirmed through FACS analysis 48 h post-transduction with a minimum efficiency of 95% tomato fluorescence. CRBN and RB1 knockout was confirmed through Western blot analysis. CRBN-targeting sgRNAs are as follow: 5′-GTCCTGCTGATCTCCTTCGC-3′ and 5′-GGATTCACATAAGCTGCCAT-3′; RB1-targeting sgRNA is as follow: 5′-GGTTCTTTGAGCAACATGGG-3′.

**Reagents and antibodies**. Lenalidomide, pomalidomide, palbociclib, melphalan, bortezomib and dexamethasone were obtained from SelleckChem. BSJ-03-123, YKL-06-102 and CST528 were synthesized according to Brand et al. 2019 and Steinebach et al. 2020.

Primary antibodies used for Western blotting from Cell Signaling (Danvers, USA) include CDK6 (clone DCS83, #3136, RRID:AB_2229289, 1:2000), CDK4 (clone D9G3E, #12790, RRID:AB_2631166, 1:1000), Rb (clone 4H1, #9309, RRID:AB_823629, 1:1000), Phospho-Rb (Ser807/811) (#9308, RRID:AB_331472, 1:1000), IKZF3 (clone D1C1E, #15103, RRID:AB_2744524, 1:1000), IKZF1 (clone D6N9Y, #14859, RRID:AB_2744523, 1:1000), IRF4 (clone D43H10, #4299, RRID:AB_10547141, 1:1000), c-Myc (clone D84C12, #5605, RRID:AB_1903938, 1:1000), RRM1 (clone D12F12, #8637, RRID:AB_11217623, 1:1000), RRM2 (clone E7Y9J, #65939, 1:1000), anti-rabbit IgG HRP-linked antibody (#7074, 1:5000), anti-mouse IgG HRP-linked antibody (#7076, 1:5000); antibodies from Sigma-Aldrich (St. Louis, USA) include anti-α-Tubulin (#T5168, RRID:AB_477579, 1:7000); antibodies from Santa Cruz Biotechnology (Dallas, USA) include CDK6 (clone B-10, sc-7961, 1:1000), TRIP13 (clone A-7, sc-514285, 1:1000); antibodies from Abcam (Cambridge, UK) include anti-β-actin (ab20272, 1:10,000).

**Immunoblotting**. Cells were treated with respective drugs for 16 h and treated cells were washed and lysed in Pierce IP lysis buffer. SDS–PAGE was performed with 10–20 μg of proteins loaded per sample. Proteins were then transferred onto PVDF membranes. Blotted membranes were blocked with 5% milk in tris-buffered saline/Tween20 (TBST) for one hour, followed by three times 10 min washes in TBST. Primary antibodies were diluted in 5% BSA in TBST and incubations were performed overnight at 4 °C. After further washes, respective secondary HRP-conjugated antibodies diluted in 5% milk were incubated for one hour at room temperature. Detection of proteins on PVDF was carried out using WesternBright ECL HRP substrate or WesternBright Sirius HRP substrate (Advansta, San Jose, USA) and imaged with ChemiDoc XRS + System (Bio-Rad, Munich, Germany). Membranes were subjected to 10 min incubation with Restore™ Western Blot Stripping Buffer (Thermo Fisher Scientific, Waltham, USA) followed by TBST washes. After brief re-activation with methanol, membranes were blocked, and further probing of proteins was carried out.

*Quantitative RT-PCR*. MM.1S cells wre treated with drugs at indicated times and concentrations. mRNA from cells were extracted using Qiagen RNeasy Kit and was performed according to manufacturer's instructions. Reverse transcription PCR

(RT-PCR) were carried out using TaqMan Reverse Transcription Reagents (Thermo Fisher Scientific, Waltham, USA).

Real-time quantitative PCR (RT-qPCR) were performed with SYBR Green Master Mix (Thermo Fisher Scientific, Waltham, USA) and StepOnePlus Real-Time PCR System. qPCR standard protocol was 95 °C for 5 min, followed by 45 cycles of 95 °C for 10 s and 60 °C for 30 s. Expression fold change was calculated with $2^{-(\Delta\Delta Ct)}$.

Primer pair sequences are as follow: 5′-ACTGTTGCACTTCACATTTTCCA-3′ and 5′-TCGAGGAGATGGGATTTGACT-3′ for TRIP13; 5′-GCTGAAACAGCTGCAACCTT-3′ and 5′-ACCATGGGAGAGTGTTTGCC-3′ for RRM1; 5′-TACCTGGACTTCCTGGACGA-3′ and 5′-AAGGCAACCAGCTCCTCAAA-3′ for MCM3; 5′-CCAAGTGTCCACGTTGGATG-3′ and 5′-TGCTCCGGGTATTTCTGCTT-3′ for MCM5; 5′-AGGTAGGAGAGAGTCGTGCC-3′ and 5′-CCACTCGGCAATGTTTTCCC-3′ for PRPS2; 5′-CAATGACCCCTTCATTGACC-3′ and 5′-GACAAGCTTCCCGTTCTCAG-3′ for GAPDH.

**Cell viability assay**. Cells were seeded in 96-well or 384-well plates with respective treatments and plates were incubated at 37 °C for 96 h. Cell viability readout was measured using CellTiter-Glo® Luminescent Cell Viability Assay (Promega, Madison, USA) and measured with POLARStar Omega plate reader (BMG Lab-Tech, Ortenberg, Germany). All conditions were normalized to the dimethyl sulfoxide- (DMSO) (Sigma Aldrich, Taufkirchen, Germany) treated control. Data represents the mean ± SD of biological triplicates.

**Software and statistical analysis**. Mass spectrometry raw data was searched with MaxQuant (Version 1.6.3.3)[66]. RNA-Seq data were aligned and quantified with STAR[71]. Analysis of proteomics and RNAseq data was performed with R(4.0.3) and R studio (Version 1.3.1093) statistical software environment. Statistical analysis of omics data in R was aided by the protigy application provided by the Broad institute on GitHub (https://github.com/broadinstitute/protigy). Statistical and graphical analyses of in vitro and in vivo experiments were performed with Prism version 8 and 9.1.0 (GraphPad Software, San Diego, CA, USA). Statistical differences were analyzed by one-way ANOVA, unpaired t tests with Welch's correction and log-rank Mantel–Cox tests. P values are displayed as follows: n.s. = $P > 0.05$; $*P \leq 0.05$; $**P \leq 0.01$; $***P \leq 0.001$; $****P \leq 0.0001$. For the generation of synergy maps, SynergyFinder was used, and ZIP and Bliss reference models were utilized[75].

**Reporting summary**. Further information on research design is available in the Nature Research Reporting Summary linked to this article.

## Data availability

The mass spectrometry proteomics data and search results generated in this study have been deposited to the ProteomeXchange Consortium via the PRIDE partner repository[70] with the dataset identifier PXD021265. The human reference proteome (UP000005640) was downloaded from UniProt in 01/2017 (https://ftp.uniprot.org/pub/databases/uniprot/previous_releases/). The RNA-sequencing data generated in this study are available on Gene Expression Omnibus under accession number GSE162403. Processed proteomics data are available in Supplementary Data 1 (patient proteomics data) and Supplementary Data 3 (MM1S proteomics data); processed patient RNAseq data is available in Supplementary Data 2. The remaining data are available within the Article, Supplementary Information or Source Data file. Source data are provided with this paper.

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

## Acknowledgements

This study was supported by the Deutsche Forschungsgemeinschaft (DFG) Emmy-Noether Program Kr3886/2-1 and 2-2 to J.K. and by the German Ministry of Education and Research (BMBF), as part of the National Research Node "Mass spectrometry in Systems Medicine" (MSCorSys), under grant agreement 031L0220B to P.M. Y.L.D.N. is enrolled in the doctoral program of the Berlin School of Integrative Oncology (BSIO). S.R.B. is a DFG research scholar (St5303/1-1). U.K. was supported by Deutsche Krebshilfe grants #70114425 and #111944.

## Author contributions

Contribution: J.K. and P.M designed the study; Y.L.D.N., E.R., S.B., A.D., S.M., O.P. and M.H. performed experiments; S.B. and M.K. collected patient material; C.S. synthesized compounds; T.C. and W.W. performed and statistically analyzed mouse experiments; Y.L.D.N., E.R., S.B., A.D., C.S., S.M., O.P., M.H., M.G., H.D., L.B., U.K., P.M. and J.K. analyzed and interpreted data; Y.L.D.N., E.R., P.M. and J.K. wrote the manuscript. All authors have read and approved the manuscript.

## Funding

## Competing interests

J.K. has received fees for advisory boards from BMS, Takeda, and Janssen. All other authors declare no competing interests.
