## [Peer Review File · Nature Communications]

Proteomic profiling reveals CDK6 upregulation as a targetable resistance mechanism for lenalidomide in multiple myelomaReviewers' Comments:

Reviewer #1:

Remarks to the Author:

In this manuscript, Krönke and colleagues set out to study non-genetic factors involved in the relapse of multiple myeloma (MM) patients to IMiD treatment. This is an important goal given that (i) mutations in genes known to be required for IMiD action (such as CRBN or IKZF1/3) are only observed in a subset of patients, and (ii) that IMiDs are front-line therapies used to treat the vast majority of MM patients. To that end, the authors have chosen an integrative strategy consisting of proteomics/phosphoproteomics as well as bulk RNA-sequencing in patient samples. This led to the identification of CDK6 as a promising candidate, which the authors further carefully validated in cell lines. Exploring the potential underlying mechanism of action, the authors convincingly concluded that the effect is not solely due to RB1 phosphorylation, but that CDK6 is likely acting as a more pleiotropic, functional hub that mediates IMiD resistance upon stabilization/upregulation on protein level in a kinase-dependent manner.

The presented research is timely and potentially of high clinical interest. While a few interesting questions remain (for instance, how CDK6 levels are stabilized, as it doesn't seem to be a transcriptional upregulation), those are clearly outside the scope of the initial discovery and validation. In sum, I hence recommend the manuscript for fast publication, also considering the potential therapeutic implications.

Below are some major and minor points that should be addressed prior to publication

Major points:

1. The authors nicely show lack of effect with a kinase-dead mutant. To fully enable a faithful comparison, it is however essential that the also probe for the levels of said mutant in comparison to the ectopic expression of the WT cDNA (via immunoblot).
2. Line 132: it appears as if the probed patient samples are from different patient pre-treatment/post-relapse. This would be important to state explicitly as a reader might get the feeling that these are matched samples from the same patients.
3. The authors should quantify the observed synergy (particularly in the presented bargraphs) via an additional statistical framework, such as the Bliss synergy score. Note that the "statistically significant cell viability reduction compared to single treatments" is per se no evidence for synergy (unless in less frequently used models such as HSA)
4. Do the mentioned IMiD-insensitive cell lines (L363 and AMO-1) have lower baseline CDK6 expression levels? To that end, a CDK4 and CDK 6 Western Blot probing all employed MM cell lines would be helpful.

Minor points:

1. Line 237: I would recommend a weaker phrasing.
2. Line 264 (regarding KEAP1): I would recommend a weaker phrasing.

I agree to waive anonymity
Georg Winter

Reviewer #2:

Remarks to the Author:

In this manuscript, the authors carried out tandem mass tag proteomics, phosphoproteomics and RNA-seq using primary tumor cells from serial multiple myeloma (MM) patients, and claim that CDK6 is a major regulator of resistance to the immunomodulatory drugs (IMiDs).

Comments

1. The authors should clarify the CRBN independency of upregulation of these proteins (CDK6, RRM1, TRIP13). As described previously, MM cells from patients with disease relapsed on IMiDs may have CRBN mutation or downregulated protein expression. Since RRM1 and CDK6 are substrates of the proteasome, upregulation of these proteins may due to low E3 ligase activity of CRBN.
2. The authors state that upregulation of CDK6 and RRM1 is post-translational mechanism. Please explain/discuss about the mechanism in "Discussion".
3. Is protein expression of CDK6 and/or RRM1 altered by IMiDs treatment?
4. Is there any difference of CDK6 expression in IMiDs-sensitive vs -resistant MM cell lines? Does acquired IMiDs-resistant cell line have higher CDK6 protein expression, compared to IMiDs-sensitive parental cell line?
5. The authors state that "No effect was observed for bortezomib" when examining sensitivity of CDK6 overexpressed cells to bortezomib. Is CDK6 expression upregulated in MM.1S cells after bortezomib treatment?
6. Knockdown/knockout of CDK6 should be performed prior to CDK6 inhibitor or degrader experiments. Does CDK6 knockdown or knockout re-sensitize acquired IMiDs-resistant MM cells with high expression of CDK6 to IMiDs treatment?
7. In Fig 3 and Fig 4, the authors should perform CDK6 inhibitor and IMiDs combination treatment in parental IMiDs-sensitive vs acquired IMiDs-resistant cells with high CDK6 expression.
8. It is also informative to show synergistic effect of IMiDs and CDK6 inhibition in patient samples.
9. Efficacy of the combination treatment should be confirmed in vivo.
10. The authors identified substrates of CDK6 by phosphoproteomics; however, these studies do not delineate the main targets mediating IMiDs resistance.
11. Figure legends of Fig.1C and 1D are not matched to figures. Also, figures should be numbered in order of citation in the text

Response to the reviewer comments

Reviewer #1, expert in proteomics (Remarks to the Author):

In this manuscript, Krönke and colleagues set out to study non-genetic factors involved in the relapse of multiple myeloma (MM) patients to IMiD treatment. This is an important goal given that (i) mutations in genes known to be required for IMiD action (such as CRBN or IKZF1/3) are only observed in a subset of patients, and (ii) that IMiDs are front-line therapies used to treat the vast majority of MM patients. To that end, the authors have chosen an integrative strategy consisting of proteomics/phosphoproteomics as well as bulk RNA-sequencing in patient samples. This led to the identification of CDK6 as a promising candidate, which the authors further carefully validated in cell lines. Exploring the potential underlying mechanism of action, the authors convincingly concluded that the effect is not solely due to RB1 phosphorylation, but that CDK6 is likely acting as a more pleiotropic, functional hub that mediates IMiD resistance upon stabilization/upregulation on protein level in a kinase-dependent manner.

The presented research is timely and potentially of high clinical interest. While a few interesting questions remain (for instance, how CDK6 levels are stabilized, as it doesn't seem to be a transcriptional upregulation), those are clearly outside the scope of the initial discovery and validation. In sum, I hence recommend the manuscript for fast publication, also considering the potential therapeutic implications.

Below are some major and minor points that should be addressed prior to publication

We thank Dr. Winter for appreciating our work. We fully agree that there are many new questions that arise from our results including the post-transcriptional regulation of CDK6 that we will investigate in the future. However, as Dr. Winter states this will take considerably more time and we feel that this is beyond the scope of our current manuscript.

Major points:

1. The authors nicely show lack of effect with a kinase-dead mutant. To fully enable a faithful comparison, it is however essential that the also probe for the levels of said mutant in comparison to the ectopic expression of the WT cDNA (via immunoblot).

This is a very reasonable question. Our initially used CDK6 antibody from Cell Signaling (clone DCS83) was not able to detect the K43M mutant, likely because it was raised against amino acids 30-65. We tested several different CDK6 antibodies now and found one (Santa Cruz, clone B-10) that was raised against the full CDK6 and capable in detecting also the CDK6 K43M mutant by western blot. This revealed that the protein expression level of the retroviral vector for normal and K43M mutant CDK6 is similar (new Figure 2C).

New Figure 2C. Overexpression of CDK6 in OPM2 cells using retroviral transduction confirmed through Western blot analysis.

2. Line 132: it appears as if the probed patient samples are from different patient pre-treatment/post-relapse. This would be important to state explicitly as a reader might get the feeling that these are matched samples from the same patients.

We apologize for the confusion, the samples used for western blot confirmation of the proteomics results are from different patients in order to have an independent cohort. We specified this more clearly in the revised manuscript and made one table containing all patients in the supplement to avoid confusion. The table is included in the supplemental now (Supplemental Figure 1).

3. The authors should quantify the observed synergy (particularly in the presented bargraphs) via an additional statistical framework, such as the Bliss synergy score. Note that the “statistically significant cell viability reduction compared to single treatments” is per se no evidence for synergy (unless in less frequently used models such as HSA)

We determined synergy by calculating the ZIP synergy score and in addition calculated now the Bliss score with the program Synergy Finder (Ianevski et al. Nucleic Acid Research 2020). Both methods provided very consistent results (new Supplemental Figure 8E).

New Supplemental Figure 8E. Heatmap of synergy scores of combination treatments with palbociclib. Synergy plots, ZIP score and Bliss score calculations were generated and performed with SynergyFinder.

4. Do the mentioned IMiD-insensitive cell lines (L363 and AMO-1) have lower baseline CDK6 expression levels? To that end, a CDK4 and CDK 6 Western Blot probing all employed MM cell lines would be helpful.

This is an interesting point. We performed the suggested western blot and quantified CDK6 protein levels in all cell lines used. Indeed, the U266 which doesn't express CDK6 was neither sensitive to CDK6 inhibition or degradation nor did we observe the synergistic effects that we observed in all other tested multiple myeloma cell lines. Among CDK6-expressing cell lines,

there was no clear correlation between CDK6 levels and sensitivity to pomalidomide or CDK6 inhibition. CDK6 protein baseline levels are highly variable among cell lines and may depend on many signaling pathways. In an isogenic cell model we show that CDK6 overexpression by retroviral vectors impairs pomalidomide sensitivity. Furthermore, we treated multiple myeloma cell lines for several weeks with low concentrations of lenalidomide to generate isogenic resistant cell lines where we found that CDK6 expression was elevated. This is highly consistent with our findings in patients that lenalidomide selects for CDK6 high expressing cells that are less sensitive to lenalidomide. In the patients analyzed by proteomics also the relative expression of a relapse to its respective pretreatment sample was more relevant than the baseline level alone (Also see response to reviewer #2 comment 2) (new Supplemental Figure 7B).

New Supplemental Figure 7B. Endogenous levels of CDK6 across all multiple myeloma cell lines. Quantification of CDK6 is normalized to tubulin and to MM.1S.

Minor points:

1. Line 237: I would recommend a weaker phrasing.

We removed the respective sentence

2. Line 264 (regarding KEAP1): I would recommend a weaker phrasing.

We stated this now more carefully.

Reviewer #2, expert in multiple myeloma genomics and proteomics (Remarks to the Author):

In this manuscript, the authors carried out tandem mass tag proteomics, phosphoproteomics and RNA-seq using primary tumor cells from serial multiple myeloma (MM) patients, and claim that CDK6 is a major regulator of resistance to the immunomodulatory drugs (IMiDs).

Comments

1. The authors should clarify the CRBN independency of upregulation of these proteins (CDK6, RRM1, TRIP13). As described previously, MM cells from patients with disease relapsed on IMiDs may have CRBN mutation or downregulated protein expression. Since RRM1 and CDK6

are substrates of the proteasome, upregulation of these proteins may be due to low E3 ligase activity of CRBN.

This is a very interesting point. As the reviewer states mutation, deletion, or downregulation of CRBN and other members of the E3 ligase complex have been shown to alter IMiD sensitivity in cell lines, and such mutations have been found in resistant patients. However, these genetic events are very rare, occurring in less than 10% of resistant patients and are frequently only heterozygous and subclonal, therefore not explaining the majority of IMiD resistance (Kortüm *et al.*, Blood 2016; Gooding *et al.*, Blood 2021). In our patients, CRBN, as well as DDB1 RNA and protein levels were not different between diagnosis and relapse (new Figure 1D). In four patients analyzed also by exome sequencing (in Bohl *et al.*, Blood Advances 2021), no mutations in *CRBN* or other members of the E3 ligase were detectable.

Inactivation of CRBN by CRISPR/Cas9 in multiple myeloma cell lines did not affect expression of CDK6, RRM1, or TRIP13. In summary, we did not find a functional connection that RRM1, CDK6, and TRIP13 are regulated by CRBN. We included a new paragraph addressing this point.

New Figure 1D. Median normalized protein intensities (\log_2 TMT intensities) of CRBN in all 10 samples were plotted against their respective normalized RNA expression levels (\log_2 TPM values). Samples from the same patient are connected.

New Supplemental Figure 2C. CRISPR/Cas9-mediated knockout of CRBN and protein levels of CDK6, TRIP13, and RRM1 detected by western blot.

2. The authors state that upregulation of CDK6 and RRM1 is post-translational mechanism. Please explain/discuss about the mechanism in "Discussion".

The regulation of CDK6 is one of the main questions emerging from our work. Our new data reveal that RRM1 and TRIP13 are altered on the RNA and protein level after palbociclib treatment indicating that this is through CDK6 indirectly affecting their transcriptional regulation as it has been described previously (Kollmann *et al.*, Cancer Cell 2013). The mechanisms driving CDK6 protein expression is unclear. The low correlation of RNA and protein levels implies that it is through a post-transcriptional or post-translational mechanism. Identification of such mechanisms is clearly important, however, this will take considerable work and time and we feel that it is beyond the scope of the current work (please see also comment by reviewer 1). We discussed this in more detail now in the manuscript.

3. Is protein expression of CDK6 and/or RRM1 altered by IMiDs treatment?

We performed IMiD treatment in MM1.S cells for different time points and found that CDK6 protein level is slightly lowered by IMiDs, which is in contrast to the upregulation of CDK6 in our proteomic study as well as our acquired IMiD-resistance myeloma cell models. Our data therefore supports that CDK6 and consequently RRM1 is upregulated as a resistance mechanism in multiple myeloma and not directly induced by the treatment (new Supplemental Figure 4B).

New Supplemental Figure 4B. Treatment of MM.1S cells with 1 μ M of Pom for 24 h, 48 h, or 72 h and CDK6 protein levels by western blot.

4. Is there any difference of CDK6 expression in IMiDs-sensitive vs -resistant MM cell lines? Does acquired IMiDs-resistant cell line have higher CDK6 protein expression, compared to IMiDs-sensitive parental cell line?

This is an important point raised by the reviewer. We performed the suggested experiments. Multiple myeloma cell lines were cultured in the presence of various concentrations of lenalidomide to obtain isogenic cell lines with acquired resistance. Indeed, after 3 and 4 weeks CDK6 protein levels were found to be upregulated in cell lines with acquired resistance as compared to the parental cell line. This upregulation of CDK6 protein in lenalidomide-resistance myeloma cells is in accordance to the proteomic results of paired patient samples that acquired resistance after long-term lenalidomide treatment. (new Supplemental Figure 4A)

New Supplemental Figure 4A. CDK6 protein levels of LenR cells and respective quantification. CDK6 protein levels were normalized to respective loading controls and to treatment control.

5. The authors state that “No effect was observed for bortezomib” when examining sensitivity of CDK6 overexpressed cells to bortezomib. Is CDK6 expression upregulated in MM.1S cells after bortezomib treatment?

Thank you for this comment. We tested the effect of different concentrations of the proteasome inhibitors bortezomib and carfilzomib. No effect was observed on CDK6 protein levels (new Supplemental Figure 4C).

New Supplemental Figure 4C. Treatment of MM.1S with proteasome inhibitors at respective concentrations for 4 h.

6. Knockdown/knockout of CDK6 should be performed prior to CDK6 inhibitor or degrader experiments. Does CDK6 knockdown or knockout re-sensitize acquired IMiDs-resistant MM cells with high expression of CDK6 to IMiDs treatment?

We tried the suggested experiment using shRNA against CDK6. However, multiple myeloma cell lines expressing CDK6 shRNA through lentiviral vectors rapidly died. A readout after selection of the transduced cells and lenalidomide or pomalidomide treatment for another 4 days (what is necessary due to the slow acting of IMiDs) was therefore not possible. However, with CDK6 inhibition and CDK6 degradation, we have two methods for inactivating CDK6 that provided highly consistent results. Overexpression of CDK6 was sufficient to partially rescue multiple myeloma cells from pomalidomide and lenalidomide further supporting the role of CDK6.

7. In Fig 3 and Fig 4, the authors should perform CDK6 inhibitor and IMiDs combination treatment in parental IMiDs-sensitive vs acquired IMiDs-resistant cells with high CDK6 expression.

Thank you for proposing this experiment. We have performed the suggested combination treatment of CDK6 inhibition with IMiDs in cells with acquired CDK6 upregulation and IMiD resistant as well as in cell with induced CDK6 overexpression through retroviral vectors. In both models CDK6 inhibition led to IMiD sensitivity comparable to the parental cells (new Figure 3E-3F).

New Supplemental Figure 3E-3F. (E) Cell viability of MM.1S LenR cells and **(F)** OPM2 CDK6 OE cells upon combination treatment of Pom and Palb at 1 μ M for 96 h. Pom = pomalidomide; Palb = palbociclib.

8. It is also informative to show synergistic effect of IMiDs and CDK6 inhibition in patient samples.

We agree that primary patient samples are a preferable model system in studying cancer. We tried the requested treatments in >20 different primary patient samples obtained directly after bone marrow aspiration in short-term cultures, with and without supporting bone marrow. However, we were not able to recapitulate our *in vitro* and *in vivo* (please see point 9) observations by the IMiD or by the combination. The lack of anti-tumor effects in primary patient samples *ex vivo*, even for the well-established IMiD treatment, is likely due to the following reasons:

- 1) it is known that primary multiple myeloma cells can only be maintained for up to 48 hours *ex vivo* but do not proliferate, even in the presence of stroma cells or cytokines (which we all tried). Lenalidomide and pomalidomide are known to act through inhibition of cell growth rather than inducing apoptosis or other forms of cell death. Therefore, lenalidomide and pomalidomide likely have no effects on non-cycling primary multiple myeloma cells *in vitro*.
- 2) In addition, even in fast proliferating multiple myeloma cell lines, a measurable effect of IMiDs can be observed the earliest after 3 to 4 days. Since primary multiple myeloma cells rapidly die *ex vivo*, no reliable results can be obtained after 3 to 4 days.

9. Efficacy of the combination treatment should be confirmed *in vivo*.

For translating our findings to patients, *in vivo* experiments in animals are highly important as pointed out by the reviewer. We therefore performed the requested experiment in an MM1.S

subcutaneous xenograft model. Two weeks after injection of the multiple myeloma cells, tumors with a size of ~200 mm³ were detected and treatment with DMSO as control, pomalidomide 5 mg/kg and palbociclib 50 mg/kg and the combination pomalidomide and palbociclib given orally per gavage on a daily basis was initiated. Tumor growth was significantly delayed by each single treatment. In contrast, the combination treatment lead to a shrinkage of tumor size to undetectable levels. Treatment was stopped after two weeks and in the single treatment arms tumors grew rapidly and mice needed to be taken out of the experiment during or shortly after treatment. In contrast, tumor growth started much later in the combination treatment arm with a significantly lower tumor size on several time points and a significantly prolonged survival. In summary, this shows that the combination of pomalidomide and CDK6 inhibition is highly active *in vivo*. (new Figure 5, new Supplemental Figure 11)

New Figure 5. Combination treatment of pomalidomide and palbociclib has high therapeutic efficacy in multiple myeloma *in vivo*. (A) Experiment setup: MM.1S cells were injected into NOG mice and treatment started 19 days after injection of myeloma cells when tumors were ~0.2 cm³. Mice were treated on a daily basis with vehicle control, pomalidomide (5 mg/kg), palbociclib (50 mg/kg) or the combination p.o. for 17 days and observed until day 28. (B) Tumor growth in treated mice of monotherapy of pomalidomide and palbociclib, and

in combination. Mice were taken out of the study when tumors exceeded a size of 1.2 cm³. (C) Survival of the four groups. Statistical differences were analyzed by log-rank Mantel-Cox test. All comparisons of survival curves resulted in p-values <0.01. Data represent mean ±SD of biological replicates. Group size: n=5 for vehicle group; n=6 for pomalidomide treatment group; n=6 for palbociclib treatment group; n=6 for pomalidomide + palbociclib treatment group; n=6 for YKL-06-102 treatment group.

New Supplemental Figure 11. Combination treatment of pomalidomide and palbociclib has high therapeutic efficacy in multiple myeloma *in vivo*. Tumor size on day 9. Differences in tumor volumes were analyzed by unpaired t-tests with Welch's correction. P values are displayed as follows: * = P ≤ 0.05; ** = P ≤ 0.01; *** = P ≤ 0.001

10. The authors identified substrates of CDK6 by phosphoproteomics; however, these studies do not delineate the main targets mediating IMiDs resistance.

This is an important point raised by the reviewer. The identified phosphopeptides in MM1S cell treated with Palbociclib or a CDK6-specific PROTAC comprised the known CDK4/6 substrates RB1, RBL1, RBL2, CDKN1A, FOXM1, and JUN. In addition, we found several proteins that were deregulated in relapse samples (DNTM1, GMPS, KLHDC4, NCAPD2, NCAPH, GLYR1, NOP56), providing a potential functional link between CDK6 kinase function and CDK6 regulated protein levels.

In addition, lower phosphorylation levels were observed in palbociclib treated cells in several transcriptional regulators including RNF169 and ZBTB38, as well as known CDK6 interactors and substrates JUN and FOXM1. Our new RT-qPCR analyses revealed that several proteins in the resistance signature were also altered on the mRNA level including RRM1 and TRIP13, implying CDK6-dependent transcriptional regulation as has been previously reported.

Identifying the CDK6-associated transcription factors regulating these genes is an important question but this needs further investigations and is beyond the scope of our current work.

11. Figure legends of Fig.1C and 1D are not matched to figures. Also, figures should be numbered in order of citation in the text

Thank you for this advice. We corrected the legends and order of citation in the text.

Reviewers' Comments:

Reviewer #1:

Remarks to the Author:

In this revised version of the manuscript, the authors have clarified all my points in a sufficient manner. I congratulate them to an important finding and look forward to seeing the study in print.

Reviewer #2:

Remarks to the Author:

Revised manuscript now much improved

Reviewer #3:

Remarks to the Author:

This reviewer reviewed this manuscript from a technical point of view at the request of the editor. In general, the technical quality in proteomics-mass spectrometry is acceptable. So it would be publishable after minor revisions as shown below:

(1) Supplementary Information-Page 3: "variable modifications were set to M-oxidation and N-terminal acetylation (M)": N-term acetylation should be exclusive for protein N-term and should not be restricted to Met. MQ can automatically take into account the possibility of deleting the first Met of a protein.

(2) Regarding the Volcano plot in Fig 1B, the authors first performed a 1-sample t-test to exclude those that fell outside the 95% or 99% confidence interval (CI), and then used the Benjamini-Hochberg method to set the cutoff value based on the FDR. They should comment why the different CI and the different FDR-cutoff values were employed? Also the descriptions were not so kind for readers. "CI" should be "confidence interval". FDR setting should be described in the method section.

(3) Information on statistical software packages should be described (Perseus or other packages?), although there is a section named "Software and statistical analysis" at the end of supplementary information.

(4) Regarding LCMS replicates, it is unclear whether the authors did replicate LCMS injections per sample.

(5) Supplementary Information-Page 2: "Proteomics and phosphoproteomics sample preparation" section: This part is quite complex. This reviewer strongly recommends preparing one table to show the relationship between TMT-11plex (10-plex?) and each sample.

(6) Line 159-161: "Of the 134 phosphopeptides that passed the significance cut-off of 0.12 FDR, only 15 belonged to proteins that were also significantly regulated on the global protein level.": How many proteins were the source of these 134 phosphopeptides, and how many of them were identified by global proteome analysis? Without this information, it is impossible to evaluate the fact that there were only 15 phosphopeptides derived from proteins that were significantly regulated.

Response to reviewer #3

We highly appreciate the thoughtful and fast review of our manuscript.

(1) Supplementary Information-Page 3: “variable modifications were set to M-oxidation and N-terminal acetylation (M)”: N-term acetylation should be exclusive for protein N-term and should not be restricted to Met. MQ can automatically take into account the possibility of deleting the first Met of a protein.

We thank the reviewer for pointing out this mistake/oversight in our description of the proteomic analysis. The preset modification “protein N-term acetylation” by MaxQuant was selected as variable modification with default parameters, which includes also absence of the first Met of a protein as the reviewer stated. We can confirm that the modification was actually not restricted to methionine.

Changed statement in supplementary information page 4:

Fixed modifications were set to carbamidomethylation of C. Variable modifications were set to M-oxidation and acetylation of protein N-termini including neo protein N-terms after cleavage of first methionine.

(2) Regarding the Volcano plot in Fig 1B, the authors first performed a 1-sample t-test to exclude those that fell outside the 95% or 99% confidence interval (CI), and then used the Benjamini-Hochberg method to set the cutoff value based on the FDR. They should comment why the different CI and the different FDR-cutoff values were employed? Also the descriptions were not so kind for readers. “CI” should be “confidence interval”. FDR setting should be described in the method section.

As it can be expected from affinity enriched samples, the variability of phosphoproteome samples was higher compared to the global proteome and this effect was stronger in primary samples compared to cell culture samples. We therefore decided to apply a stronger filtering approach for outliers based on confidence intervals as well as to use a less stringent FDR threshold in the phospho-proteomic analysis of patient samples. We adjusted the description in the method section and included the FDR settings.

Changed statement in manuscript on page 5:

The processed data was analyzed with a moderated one-sample t-test (limma package²⁸) and subjected to reproducibility filtering: global proteome (99% confidence interval); phosphoproteome (95% confidence interval) to remove outliers prior to applying multiple testing correction with the Benjamini-Hochberg method. As expected, the phosphoproteome showed higher variability compared to the global proteome, which is more stable towards dynamic modifications. We therefore applied a stronger outlier filter and more relaxed cutoff for the phosphoproteomic data. A cutoff of 10% and 12% FDR was implemented for global proteome (Figure 1B) and phosphoproteome (Supplemental Figure 3C) analysis of patients samples, respectively.

(3) Information on statistical software packages should be described (Perseus or other packages?), although there is a section named “Software and statistical analysis” at the end of supplementary information.

MaxQuant output was analyzed with the R statistical software environment. Statistical analysis of mass spec data was aided by the protigy application provided by the Broad institute on GitHub (<https://github.com/broadinstitute/protigy>). Moderated one-sample t -tests were performed with the limma package integrated in the protigy application. We added the relevant information in the main text as well as in the supplementary file in the first paragraph about

data analysis. We added names of R packages and citations in the supplementary section of “Mass spectrometry data analysis” where appropriate. We also clarified this in the section named “Software and statistical analysis” at the end of supplementary information

We included the following statements:

Supplementary information page 4:

Analysis of MaxQuant output was performed with R (4.0.3) and R studio (Version 1.3.1093) statistical software environment. Statistical analysis in R was aided by the protigy application provided by the Broad institute on GitHub (<https://github.com/broadinstitute/protigy>) as used previously (Udeshi et al. 2020).

Supplementary information page 4:

The processed data was analyzed with a moderated one-sample t-test (limma package)

Supplementary information page 4:

Software and statistical analysis

*Mass spectrometry raw data was searched with MaxQuant (Version 1.6.3.3)⁴. RNA-Seq data were aligned and quantified by using STAR as previously described.⁶ Analysis of proteomics and RNAseq data was performed with R(4.0.3) and R studio (Version 1.3.1093) statistical software environment. Statistical analysis of omics data in R was aided by the protigy application provided by the Broad institute on GitHub (<https://github.com/broadinstitute/protigy>). Statistical and graphical analyses of in vitro and in vivo experiments were performed with Prism version 8 and 9.1.0 (GraphPad Software, San Diego, CA, USA). Statistical differences were analyzed by one-way ANOVA, unpaired t tests with Welch’s correction and log-rank Mantel-Cox tests. P values are displayed as follows: n.s. = $P > 0.05$; * = $P \leq 0.05$; ** = $P \leq 0.01$; *** = $P \leq 0.001$; **** = $P \leq 0.0001$. For the generation of synergy maps, SynergyFinder was used, and ZIP and Bliss reference models were utilized.*

(4) Regarding LCMS replicates, it is unclear whether the authors did replicate LCMS injections per sample.

Combined TMT labelled samples were fractionated off-line into 24 fractions using high-pH reversed phase chromatography. Each fraction was injected once (no replicate injections). We clarified this in the detailed method section of the supplementary file.

We included the following statement in the supplementary information on page 3:

Mass spectrometry raw data was acquired on a Q-Exactive HF-X connected to an EASY-nLC 1200system (both Thermo Fisher Scientific). Each HpH fraction was measured individually as a single LCMS injection.

(5) Supplementary Information-Page 2:”Proteomics and phosphoproteomics sample preparation” section: This part is quite complex. This reviewer strongly recommends preparing one table to show the relationship between TMT-11plex (10-plex?) and each sample.

Thank you for pointing this out. We modified the sample preparation section in the supplementary information on page 2 to improve clarity. Channel annotation and sample descriptions were added in the supplementary tables containing proteomics data of patients (Supplementary Table 1) and MM1S cells (Supplementary Table S3). We indicated where to find channel annotation in the text

Changed sample preparation section on page 2 of the supplementary information:

Proteomics and phosphoproteomics sample preparation

Samples for global proteome and phospho-proteome profiling were processed as described previously.² In brief, samples were lysed at 4 °C with urea lysis buffer (8 M urea, 50mM Tris (pH 8), 150mM NaCl) supplemented with protease inhibitors (2 µg/ml aprotinin, 10 µg/ml leupeptin 1mM phenylmethylsulfonylflourid) and phosphatase inhibitors (10mM NaF, phosphatase inhibitor cocktail 1 and 2, Sigma Aldrich). Extracted proteins were reduced with 5mM dithiothreitol for 1 h and alkylated with 10mM iodoacetamide for 45 min in the dark. Sequencing grade LysC (Wako) was added at a ratio of 1:50 and after 2 h, samples were diluted 1:4 with 50mM Tris-HCl Ph8 and sequencing grade trypsin (Promega) was added at a ratio of 1:50. Digestion was completed overnight and subsequently acidified samples were desalted with Sep-Pak C18 cc Cartridges (Waters). Dried samples were resuspended in 10mM HEPES (pH 8.5) and peptide concentration was determined. For relative quantification of proteins in multiple myeloma patient samples, 10 µg peptides for each sample were labeled with 10-plexing tandem mass tags (TMT-10; Thermo Fisher Scientific) and combined into one TMT plex. For quantification of phosphopeptides in multiple myeloma patient samples, 150 µg of peptides for the samples 1 to 9 and 23 µg of starting material for sample 10 were labeled with TMT-10 reagents and combined into a second plex. **TMT-channel annotation of patient proteomic data is available in Supplementary Table 1.** For global and phosphoproteome analysis of MM.1S drug-treated cells, 200 µg peptides of each sample were labeled with 11-plexing tandem mass tags (TMT-11; Thermo Fisher Scientific) and the samples were randomly distributed into two TMT plexes. For the MM.1S experiment, an internal reference sample composed of equal amounts of peptide material from all samples was included in each TMT 11-plex to provide a standard for relative quantification. **TMT-channel annotation of MM.1S proteomic data is available in Supplementary Table 3.** Combined TMT samples were fractionated off-line into 24 fractions using high-pH reversed phase chromatography. For analysis of the global proteome, 500 ng peptides of each fraction were analyzed with LC-MS/MS. The remaining material was pooled into 12 fractions for each plex and subjected to phosphopeptide enrichment with immobilized metal affinity chromatography (IMAC) automated on an AssayMap Bravo System (Agilent).³

(6) Line 159-161: “Of the 134 phosphopeptides that passed the significance cut-off of 0.12 FDR, only 15 belonged to proteins that were also significantly regulated on the global protein level.”: How many proteins were the source of these 134 phosphopeptides, and how many of them were identified by global proteome analysis? Without this information, it is impossible to evaluate the fact that there were only 15 phosphopeptides derived from proteins that were significantly regulated.

To improve clarity, we changed the corresponding text in the manuscript on page 8 as outlined below:

We detected 24,797 phosphopeptides derived from 5,698 proteins (Supplemental Figure 3C, Supplemental Table 1). In total, 134 phosphopeptides passed the 0.12 FDR significance cutoff. The majority of the proteins, that the significant phosphopeptides originated from, were also detected in the global proteome analysis of patient samples (92 out of 112). However, only 15 of the significant phosphopeptides belonged to proteins that were also significantly regulated on the global protein level.

Reviewers' Comments:

Reviewer #3:

Remarks to the Author:

This is a revised version of NCOMMS-21-02753A. The authors have revised the manuscript promptly and this reviewer has no more comment.